# Policy Gradient for Coherent Risk Measures

**Aviv Tamar**
UC Berkeley
avivt@berkeley.edu

**Yinlam Chow**
Stanford University
ychow@stanford.edu

**Mohammad Ghavamzadeh**
Adobe Research & INRIA
mohammad.ghavamzadeh@inria.fr

**Shie Mannor**
Technion
shie@ee.technion.ac.il

## Abstract

Several authors have recently developed risk-sensitive policy gradient methods that augment the standard expected cost minimization problem with a measure of *variability* in cost. These studies have focused on *specific* risk-measures, such as the variance or conditional value at risk (CVaR). In this work, we extend the policy gradient method to *the whole class* of coherent risk measures, which is widely accepted in finance and operations research, among other fields. We consider both static and time-consistent dynamic risk measures. For static risk measures, our approach is in the spirit of *policy gradient* algorithms and combines a standard sampling approach with convex programming. For dynamic risk measures, our approach is *actor-critic* style and involves explicit approximation of value function. Most importantly, our contribution presents a *unified* approach to risk-sensitive reinforcement learning that generalizes and extends previous results.

## 1 Introduction

Risk-sensitive optimization considers problems in which the objective involves a *risk measure* of the random cost, in contrast to the typical *expected* cost objective. Such problems are important when the decision-maker wishes to manage the *variability* of the cost, in addition to its expected outcome, and are standard in various applications of finance and operations research. In reinforcement learning (RL) [27], risk-sensitive objectives have gained popularity as a means to regularize the variability of the total (discounted) cost/reward in a Markov decision process (MDP).

Many risk objectives have been investigated in the literature and applied to RL, such as the celebrated Markowitz mean-variance model [16], Value-at-Risk (VaR) and Conditional Value at Risk (CVaR) [18, 29, 21, 10, 8, 30]. The view taken in this paper is that the preference of one risk measure over another is *problem-dependent* and depends on factors such as the cost distribution, sensitivity to rare events, ease of estimation from data, and computational tractability of the optimization problem. However, the highly influential paper of Artzner et al. [2] identified a set of natural properties that are desirable for a risk measure to satisfy. Risk measures that satisfy these properties are termed *coherent* and have obtained widespread acceptance in financial applications, among others. We focus on such coherent measures of risk in this work.

For sequential decision problems, such as MDPs, another desirable property of a risk measure is *time consistency*. A time-consistent risk measure satisfies a "dynamic programming" style property: if a strategy is risk-optimal for an $n$-stage problem, then the component of the policy from the $t$-th time until the end (where $t < n$) is also risk-optimal (see principle of optimality in [5]). The recently proposed class of dynamic Markov coherent risk measures [24] satisfies both the coherence and time consistency properties.

In this work, we present policy gradient algorithms for RL with a coherent risk objective. Our approach applies to *the whole class* of coherent risk measures, thereby generalizing and unifying previous approaches that have focused on individual risk measures. We consider both *static* coherent

risk of the total discounted return from an MDP and time-consistent *dynamic* Markov coherent risk. Our main contribution is formulating the risk-sensitive policy-gradient under the coherent-risk framework. More specifically, we provide:

- A new formula for the gradient of static coherent risk that is convenient for approximation using sampling.

- An algorithm for the gradient of general static coherent risk that involves sampling with convex programming and a corresponding consistency result.

- A new policy gradient theorem for Markov coherent risk, relating the gradient to a suitable *value function* and a corresponding actor-critic algorithm.

Several previous results are special cases of the results presented here; our approach allows to re-derive them in greater generality and simplicity.

**Related Work**   Risk-sensitive optimization in RL for specific risk functions has been studied recently by several authors. [6] studied exponential utility functions, [18], [29], [21] studied mean-variance models, [8], [30] studied CVaR in the static setting, and [20], [9] studied dynamic coherent risk for systems with linear dynamics. Our paper presents a general method *for the whole class* of coherent risk measures (both static and dynamic) and is not limited to a specific choice within that class, nor to particular system dynamics.

Reference [19] showed that an MDP with a dynamic coherent risk objective is essentially a robust MDP. The planning for large scale MDPs was considered in [31], using an approximation of the value function. For many problems, approximation in the policy space is more suitable (see, e.g., [15]). Our sampling-based RL-style approach is suitable for approximations both in the policy and value function, and scales-up to large or continuous MDPs. We do, however, make use of a technique of [31] in a part of our method.

Optimization of coherent risk measures was thoroughly investigated by Ruszczynski and Shapiro [25] (see also [26]) for the stochastic programming case in which the policy parameters do not affect the distribution of the stochastic system (i.e., the MDP trajectory), but only the reward function, and thus, this approach is not suitable for most RL problems. For the case of MDPs and dynamic risk, [24] proposed a dynamic programming approach. This approach does not scale-up to large MDPs, due to the "curse of dimensionality". For further motivation of risk-sensitive policy gradient methods, we refer the reader to [18, 29, 21, 8, 30].

## 2   Preliminaries

Consider a probability space $(\Omega, \mathcal{F}, P_\theta)$, where $\Omega$ is the set of outcomes (sample space), $\mathcal{F}$ is a $\sigma$-algebra over $\Omega$ representing the set of events we are interested in, and $P_\theta \in \mathcal{B}$, where $\mathcal{B} := \left\{\xi : \int_{\omega \in \Omega} \xi(\omega) = 1, \xi \geq 0\right\}$ is the set of probability distributions, is a probability measure over $\mathcal{F}$ parameterized by some tunable parameter $\theta \in \mathbb{R}^K$. In the following, we suppress the notation of $\theta$ in $\theta$-dependent quantities.

To ease the technical exposition, in this paper we restrict our attention to finite probability spaces, i.e., $\Omega$ has a finite number of elements. Our results can be extended to the $L_p$-normed spaces without loss of generality, but the details are omitted for brevity.

Denote by $\mathcal{Z}$ the space of random variables $Z : \Omega \mapsto (-\infty, \infty)$ defined over the probability space $(\Omega, \mathcal{F}, P_\theta)$. In this paper, a random variable $Z \in \mathcal{Z}$ is interpreted as a cost, i.e., the smaller the realization of $Z$, the better. For $Z, W \in \mathcal{Z}$, we denote by $Z \leq W$ the point-wise partial order, i.e., $Z(\omega) \leq W(\omega)$ for all $\omega \in \Omega$. We denote by $\mathbb{E}_\xi[Z] \doteq \sum_{\omega \in \Omega} P_\theta(\omega)\xi(\omega)Z(\omega)$ a $\xi$-weighted expectation of $Z$.

An MDP is a tuple $\mathcal{M} = (\mathcal{X}, \mathcal{A}, C, P, \gamma, x_0)$, where $\mathcal{X}$ and $\mathcal{A}$ are the state and action spaces; $C(x) \in [-C_{\max}, C_{\max}]$ is a bounded, deterministic, and state-dependent cost; $P(\cdot|x, a)$ is the transition probability distribution; $\gamma$ is a discount factor; and $x_0$ is the initial state.[1] Actions are chosen according to a $\theta$-parameterized stationary Markov[2] policy $\mu_\theta(\cdot|x)$. We denote by $x_0, a_0, \ldots, x_T, a_T$ a trajectory of length $T$ drawn by following the policy $\mu_\theta$ in the MDP.

## 2.1 Coherent Risk Measures

A *risk measure* is a function $\rho : \mathcal{Z} \to \mathbb{R}$ that maps an uncertain outcome $Z$ to the extended real line $\mathbb{R} \cup \{+\infty, -\infty\}$, e.g., the expectation $\mathbb{E}[Z]$ or the conditional value-at-risk (CVaR) $\min_{\nu \in \mathbb{R}} \{\nu + \frac{1}{\alpha}\mathbb{E}[(Z - \nu)^+]\}$. A risk measure is called *coherent*, if it satisfies the following conditions for all $Z, W \in \mathcal{Z}$ [2]:

**A1** Convexity: $\forall \lambda \in [0, 1], \ \rho(\lambda Z + (1 - \lambda)W) \leq \lambda \rho(Z) + (1 - \lambda)\rho(W)$;

**A2** Monotonicity: if $Z \leq W$, then $\rho(Z) \leq \rho(W)$;

**A3** Translation invariance: $\forall a \in \mathbb{R}, \ \rho(Z + a) = \rho(Z) + a$;

**A4** Positive homogeneity: if $\lambda \geq 0$, then $\rho(\lambda Z) = \lambda \rho(Z)$.

Intuitively, these condition ensure the "rationality" of single-period risk assessments: A1 ensures that diversifying an investment will reduce its risk; A2 guarantees that an asset with a higher cost for every possible scenario is indeed riskier; A3, also known as 'cash invariance', means that the deterministic part of an investment portfolio does not contribute to its risk; the intuition behind A4 is that doubling a position in an asset doubles its risk. We further refer the reader to [2] for a more detailed motivation of coherent risk.

The following representation theorem [26] shows an important property of coherent risk measures that is fundamental to our gradient-based approach.

**Theorem 2.1.** *A risk measure $\rho : \mathcal{Z} \to \mathbb{R}$ is coherent if and only if there exists a convex bounded and closed set $\mathcal{U} \subset \mathcal{B}$ such that*[3]

$$\rho(Z) = \max_{\xi \,:\, \xi P_\theta \in \mathcal{U}(P_\theta)} \mathbb{E}_\xi[Z]. \tag{1}$$

The result essentially states that any coherent risk measure is an expectation w.r.t. a worst-case density function $\xi P_\theta$, i.e., a re-weighting of $P_\theta$ by $\xi$, chosen adversarially from a suitable set of test density functions $\mathcal{U}(P_\theta)$, referred to as *risk envelope*. Moreover, a coherent risk measure is *uniquely represented* by its risk envelope. In the sequel, we shall interchangeably refer to coherent risk measures either by their explicit functional representation, or by their corresponding risk-envelope.

In this paper, we assume that the risk envelope $\mathcal{U}(P_\theta)$ is given in a canonical convex programming formulation, and satisfies the following conditions.

**Assumption 2.2** (The General Form of Risk Envelope). *For each given policy parameter $\theta \in \mathbb{R}^K$, the risk envelope $\mathcal{U}$ of a coherent risk measure can be written as*

$$\mathcal{U}(P_\theta) = \left\{ \xi P_\theta : \ g_e(\xi, P_\theta) = 0, \ \forall e \in \mathcal{E}, \ f_i(\xi, P_\theta) \leq 0, \ \forall i \in \mathcal{I}, \ \sum_{\omega \in \Omega} \xi(\omega)P_\theta(\omega) = 1, \ \xi(\omega) \geq 0 \right\}, \tag{2}$$

*where each constraint $g_e(\xi, P_\theta)$ is an affine function in $\xi$, each constraint $f_i(\xi, P_\theta)$ is a convex function in $\xi$, and there exists a strictly feasible point $\bar{\xi}$. $\mathcal{E}$ and $\mathcal{I}$ here denote the sets of equality and inequality constraints, respectively. Furthermore, for any given $\xi \in \mathcal{B}$, $f_i(\xi, p)$ and $g_e(\xi, p)$ are twice differentiable in $p$, and there exists a $M > 0$ such that*

$$\max \left\{ \max_{i \in \mathcal{I}} \left| \frac{df_i(\xi, p)}{dp(\omega)} \right|, \max_{e \in \mathcal{E}} \left| \frac{dg_e(\xi, p)}{dp(\omega)} \right| \right\} \leq M, \ \forall \omega \in \Omega.$$

Assumption 2.2 implies that the risk envelope $\mathcal{U}(P_\theta)$ is known in an *explicit* form. From Theorem 6.6 of [26], in the case of a finite probability space, $\rho$ is a coherent risk if and only if $\mathcal{U}(P_\theta)$ is a convex and compact set. This justifies the affine assumption of $g_e$ and the convex assumption of $f_i$. Moreover, the additional assumption on the smoothness of the constraints holds for many popular coherent risk measures, such as the CVaR, the mean-semi-deviation, and spectral risk measures [1].

## 2.2 Dynamic Risk Measures

The risk measures defined above do not take into account any temporal structure that the random variable might have, such as when it is associated with the return of a trajectory in the case of MDPs. In this sense, such risk measures are called *static*. *Dynamic* risk measures, on the other hand,

---

policies on a state space augmented with accumulated cost. The latter has shown to be sufficient for optimizing the CVaR risk [4].

[3]When we study risk in MDPs, the risk envelope $\mathcal{U}(P_\theta)$ in Eq. 1 also depends on the state $x$.

explicitly take into account the temporal nature of the stochastic outcome. A primary motivation for considering such measures is the issue of *time consistency*, usually defined as follows [24]: if a certain outcome is considered less risky in all states of the world at stage $t + 1$, then it should also be considered less risky at stage $t$. Example 2.1 in [13] shows the importance of time consistency in the evaluation of risk in a dynamic setting. It illustrates that for multi-period decision-making, optimizing a static measure can lead to "time-inconsistent" behavior. Similar paradoxical results could be obtained with other risk metrics; we refer the readers to [24] and [13] for further insights.

**Markov Coherent Risk Measures.** Markov risk measures were introduced in [24] and constitute a useful class of dynamic time-consistent risk measures that are important to our study of risk in MDPs. For a $T$-length horizon and MDP $\mathcal{M}$, the Markov coherent risk measure $\rho_T(\mathcal{M})$ is

$$\rho_T(\mathcal{M}) = C(x_0) + \gamma\rho\bigg(C(x_1) + \ldots + \gamma\rho\Big(C(x_{T-1}) + \gamma\rho\big(C(x_T)\big)\Big)\bigg), \tag{3}$$

where $\rho$ is a static coherent risk measure that satisfies Assumption 2.2 and $x_0, \ldots, x_T$ is a trajectory drawn from the MDP $\mathcal{M}$ under policy $\mu_\theta$. It is important to note that in (3), each static coherent risk $\rho$ at state $x \in \mathcal{X}$ is induced by the transition probability $P_\theta(\cdot|x) = \sum_{a \in \mathcal{A}} P(x'|x, a)\mu_\theta(a|x)$. We also define $\rho_\infty(\mathcal{M}) \doteq \lim_{T \to \infty} \rho_T(\mathcal{M})$, which is well-defined since $\gamma < 1$ and the cost is bounded. We further assume that $\rho$ in (3) is a *Markov risk* measure, i.e., the evaluation of each static coherent risk measure $\rho$ is not allowed to depend on the whole past.

## 3 Problem Formulation

In this paper, we are interested in solving two risk-sensitive optimization problems. Given a random variable $Z$ and a static coherent risk measure $\rho$ as defined in Section 2, the static risk problem (SRP) is given by

$$\min_\theta \quad \rho(Z). \tag{4}$$

For example, in an RL setting, $Z$ may correspond to the cumulative discounted cost $Z = C(x_0) + \gamma C(x_1) + \cdots + \gamma^T C(x_T)$ of a trajectory induced by an MDP with a policy parameterized by $\theta$.

For an MDP $\mathcal{M}$ and a dynamic Markov coherent risk measure $\rho_T$ as defined by Eq. 3, the dynamic risk problem (DRP) is given by

$$\min_\theta \quad \rho_\infty(\mathcal{M}). \tag{5}$$

Except for very limited cases, there is no reason to hope that neither the SRP in (4) nor the DRP in (5) should be tractable problems, since the dependence of the risk measure on $\theta$ may be complex and non-convex. In this work, we aim towards a more modest goal and search for a *locally* optimal $\theta$. Thus, the main problem that we are trying to solve in this paper is how to calculate the gradients of the SRP's and DRP's objective functions

$$\nabla_\theta \rho(Z) \qquad \text{and} \qquad \nabla_\theta \rho_\infty(\mathcal{M}).$$

We are interested in non-trivial cases in which the gradients cannot be calculated analytically. In the static case, this would correspond to a non-trivial dependence of $Z$ on $\theta$. For dynamic risk, we also consider cases where the state space is too large for a tractable computation. Our approach for dealing with such difficult cases is through sampling. We assume that in the static case, we may obtain i.i.d. samples of the random variable $Z$. For the dynamic case, we assume that for each state and action $(x, a)$ of the MDP, we may obtain i.i.d. samples of the next state $x' \sim P(\cdot|x, a)$. We show that sampling may indeed be used in both cases to devise suitable estimators for the gradients.

To finally solve the SRP and DRP problems, a gradient estimate may be plugged into a standard stochastic gradient descent (SGD) algorithm for learning a locally optimal solution to (4) and (5). From the structure of the dynamic risk in Eq. 3, one may think that a gradient estimator for $\rho(Z)$ may help us to estimate the gradient $\nabla_\theta \rho_\infty(\mathcal{M})$. Indeed, we follow this idea and begin with estimating the gradient in the static risk case.

## 4 Gradient Formula for Static Risk

In this section, we consider a static coherent risk measure $\rho(Z)$ and propose sampling-based estimators for $\nabla_\theta \rho(Z)$. We make the following assumption on the policy parametrization, which is standard in the policy gradient literature [15].

**Assumption 4.1.** *The likelihood ratio $\nabla_\theta \log P(\omega)$ is well-defined and bounded for all $\omega \in \Omega$.*

Moreover, our approach implicitly assumes that given some $\omega \in \Omega$, $\nabla_\theta \log P(\omega)$ may be easily calculated. This is also a standard requirement for policy gradient algorithms [15] and is satisfied in various applications such as queueing systems, inventory management, and financial engineering (see, e.g., the survey by Fu [11]).

Using Theorem 2.1 and Assumption 2.2, for each $\theta$, we have that $\rho(Z)$ is the solution to the convex optimization problem (1) (for that value of $\theta$). The Lagrangian function of (1), denoted by $L_\theta(\xi, \lambda^\mathcal{P}, \lambda^\mathcal{E}, \lambda^\mathcal{I})$, may be written as

$$L_\theta(\xi, \lambda^\mathcal{P}, \lambda^\mathcal{E}, \lambda^\mathcal{I}) = \sum_{\omega \in \Omega} \xi(\omega) P_\theta(\omega) Z(\omega) - \lambda^\mathcal{P} \left( \sum_{\omega \in \Omega} \xi(\omega) P_\theta(\omega) - 1 \right) - \sum_{e \in \mathcal{E}} \lambda^\mathcal{E}(e) g_e(\xi, P_\theta) - \sum_{i \in \mathcal{I}} \lambda^\mathcal{I}(i) f_i(\xi, P_\theta).$$

(6)

The convexity of (1) and its strict feasibility due to Assumption 2.2 implies that $L_\theta(\xi, \lambda^\mathcal{P}, \lambda^\mathcal{E}, \lambda^\mathcal{I})$ has a non-empty set of saddle points $\mathcal{S}$. The next theorem presents a formula for the gradient $\nabla_\theta \rho(Z)$. As we shall subsequently show, this formula is particularly convenient for devising sampling based estimators for $\nabla_\theta \rho(Z)$.

**Theorem 4.2.** *Let Assumptions 2.2 and 4.1 hold. For any saddle point $(\xi_\theta^*, \lambda_\theta^{*,\mathcal{P}}, \lambda_\theta^{*,\mathcal{E}}, \lambda_\theta^{*,\mathcal{I}}) \in \mathcal{S}$ of* (6)*, we have*

$$\nabla_\theta \rho(Z) = \mathbb{E}_{\xi_\theta^*} \left[ \nabla_\theta \log P(\omega)(Z - \lambda_\theta^{*,\mathcal{P}}) \right] - \sum_{e \in \mathcal{E}} \lambda_\theta^{*,\mathcal{E}}(e) \nabla_\theta g_e(\xi_\theta^*; P_\theta) - \sum_{i \in \mathcal{I}} \lambda_\theta^{*,\mathcal{I}}(i) \nabla_\theta f_i(\xi_\theta^*; P_\theta).$$

The proof of this theorem, given in the supplementary material, involves an application of the Envelope theorem [17] and a standard 'likelihood-ratio' trick. We now demonstrate the utility of Theorem 4.2 with several examples in which we show that it generalizes previously known results, and also enables deriving new useful gradient formulas.

## 4.1 Example 1: CVaR

The CVaR at level $\alpha \in [0, 1]$ of a random variable $Z$, denoted by $\rho_{\text{CVaR}}(Z; \alpha)$, is a very popular coherent risk measure [23], defined as

$$\rho_{\text{CVaR}}(Z; \alpha) \doteq \inf_{t \in \mathbb{R}} \left\{ t + \alpha^{-1} \mathbb{E} \left[ (Z - t)_+ \right] \right\}.$$

When $Z$ is continuous, $\rho_{\text{CVaR}}(Z; \alpha)$ is well-known to be the mean of the $\alpha$-tail distribution of $Z$, $\mathbb{E}[Z \mid Z > q_\alpha]$, where $q_\alpha$ is a $(1 - \alpha)$-quantile of $Z$. Thus, selecting a small $\alpha$ makes CVaR particularly sensitive to rare, but very high costs.

The risk envelope for CVaR is known to be [26] $\mathcal{U} = \{\xi P_\theta : \xi(\omega) \in [0, \alpha^{-1}], \sum_{\omega \in \Omega} \xi(\omega) P_\theta(\omega) = 1\}$. Furthermore, [26] show that the saddle points of (6) satisfy $\xi_\theta^*(\omega) = \alpha^{-1}$ when $Z(\omega) > \lambda_\theta^{*,\mathcal{P}}$, and $\xi_\theta^*(\omega) = 0$ when $Z(\omega) < \lambda_\theta^{*,\mathcal{P}}$, where $\lambda_\theta^{*,\mathcal{P}}$ is any $(1 - \alpha)$-quantile of $Z$. Plugging this result into Theorem 4.2, we can easily show that

$$\nabla_\theta \rho_{\text{CVaR}}(Z; \alpha) = \mathbb{E} \left[ \nabla_\theta \log P(\omega)(Z - q_\alpha) \mid Z(\omega) > q_\alpha \right].$$

This formula was recently proved in [30] for the case of continuous distributions by an explicit calculation of the conditional expectation, and under several additional smoothness assumptions. Here we show that it holds regardless of these assumptions and in the discrete case as well. Our proof is also considerably simpler.

## 4.2 Example 2: Mean-Semideviation

The semi-deviation of a random variable $Z$ is defined as $\mathbb{SD}[Z] \doteq \left( \mathbb{E} \left[ (Z - \mathbb{E}[Z])_+^2 \right] \right)^{1/2}$. The semi-deviation captures the variation of the cost only *above its mean*, and is an appealing alternative to the standard deviation, which does not distinguish between the variability of upside and downside deviations. For some $\alpha \in [0, 1]$, the *mean-semideviation* risk measure is defined as $\rho_{\text{MSD}}(Z; \alpha) \doteq \mathbb{E}[Z] + \alpha \mathbb{SD}[Z]$, and is a coherent risk measure [26]. We have the following result:

**Proposition 4.3.** *Under Assumption 4.1, with $\nabla_\theta \mathbb{E}[Z] = \mathbb{E}[\nabla_\theta \log P(\omega) Z]$, we have*

$$\nabla_\theta \rho_{MSD}(Z; \alpha) = \nabla_\theta \mathbb{E}[Z] + \frac{\alpha \mathbb{E}\left[ (Z - \mathbb{E}[Z])_+ (\nabla_\theta \log P(\omega)(Z - \mathbb{E}[Z]) - \nabla_\theta \mathbb{E}[Z]) \right]}{\mathbb{SD}(Z)}.$$

This proposition can be used to devise a sampling based estimator for $\nabla_\theta \rho_{\text{MSD}}(Z; \alpha)$ by replacing all the expectations with sample averages. The algorithm along with the proof of the proposition are in the supplementary material. In Section 6 we provide a numerical illustration of optimization with a mean-semideviation objective.

### 4.3 General Gradient Estimation Algorithm

In the two previous examples, we obtained a gradient formula by *analytically* calculating the Lagrangian saddle point (6) and plugging it into the formula of Theorem 4.2. We now consider a general coherent risk $\rho(Z)$ for which, in contrast to the CVaR and mean-semideviation cases, the Lagrangian saddle-point is not known analytically. *We only assume that we know the structure of the risk-envelope* as given by (2). We show that in this case, $\nabla_\theta \rho(Z)$ may be estimated using a *sample average approximation* (SAA; [26]) of the formula in Theorem 4.2.

Assume that we are given $N$ i.i.d. samples $\omega_i \sim P_\theta$, $i = 1, \dots, N$, and let $P_{\theta;N}(\omega) \doteq \frac{1}{N} \sum_{i=1}^{N} \mathbb{I}\{\omega_i = \omega\}$ denote the corresponding empirical distribution. Also, let the *sample risk envelope* $\mathcal{U}(P_{\theta;N})$ be defined according to Eq. 2 with $P_\theta$ replaced by $P_{\theta;N}$. Consider the following SAA version of the optimization in Eq. 1:

$$\rho_N(Z) = \max_{\xi: \xi P_{\theta;N} \in \mathcal{U}(P_{\theta;N})} \sum_{i \in 1, \dots, N} P_{\theta;N}(\omega_i) \xi(\omega_i) Z(\omega_i). \tag{7}$$

Note that (7) defines a convex optimization problem with $\mathcal{O}(N)$ variables and constraints. In the following, we assume that a solution to (7) may be computed efficiently using standard convex programming tools such as interior point methods [7]. Let $\xi_{\theta;N}^*$ denote a solution to (7) and $\lambda_{\theta;N}^{*,\mathcal{P}}, \lambda_{\theta;N}^{*,\mathcal{E}}, \lambda_{\theta;N}^{*,\mathcal{I}}$ denote the corresponding KKT multipliers, which can be obtained from the convex programming algorithm [7]. We propose the following estimator for the gradient-based on Theorem 4.2:

$$\nabla_{\theta;N} \rho(Z) = \sum_{i=1}^{N} P_{\theta;N}(\omega_i) \xi_{\theta;N}^*(\omega_i) \nabla_\theta \log P(\omega_i)(Z(\omega_i) - \lambda_{\theta;N}^{*,\mathcal{P}}) \tag{8}$$

$$- \sum_{e \in \mathcal{E}} \lambda_{\theta;N}^{*,\mathcal{E}}(e) \nabla_\theta g_e(\xi_{\theta;N}^*; P_{\theta;N}) - \sum_{i \in \mathcal{I}} \lambda_{\theta;N}^{*,\mathcal{I}}(i) \nabla_\theta f_i(\xi_{\theta;N}^*; P_{\theta;N}).$$

Thus, our gradient estimation algorithm is a two-step procedure involving *both sampling and convex programming*. In the following, we show that under some conditions on the set $\mathcal{U}(P_\theta)$, $\nabla_{\theta;N} \rho(Z)$ is a consistent estimator of $\nabla_\theta \rho(Z)$. The proof has been reported in the supplementary material.

**Proposition 4.4.** *Let Assumptions 2.2 and 4.1 hold. Suppose there exists a compact set $C = C_\xi \times C_\lambda$ such that: (I) The set of Lagrangian saddle points $\mathcal{S} \subset C$ is non-empty and bounded. (II) The functions $f_e(\xi, P_\theta)$ for all $e \in \mathcal{E}$ and $f_i(\xi, P_\theta)$ for all $i \in \mathcal{I}$ are finite-valued and continuous (in $\xi$) on $C_\xi$. (III) For $N$ large enough, the set $\mathcal{S}_N$ is non-empty and $\mathcal{S}_N \subset C$ w.p. 1. Further assume that: (IV) If $\xi_N P_{\theta;N} \in \mathcal{U}(P_{\theta;N})$ and $\xi_N$ converges w.p. 1 to a point $\xi$, then $\xi P_\theta \in \mathcal{U}(P_\theta)$. We then have that $\lim_{N \to \infty} \rho_N(Z) = \rho(Z)$ and $\lim_{N \to \infty} \nabla_{\theta;N} \rho(Z) = \nabla_\theta \rho(Z)$ w.p. 1.*

The set of assumptions for Proposition 4.4 is large, but rather mild. Note that (I) is implied by the Slater condition of Assumption 2.2. For satisfying (III), we need that the risk be well-defined for every empirical distribution, which is a natural requirement. Since $P_{\theta;N}$ always converges to $P_\theta$ uniformly on $\Omega$, (IV) essentially requires smoothness of the constraints. We remark that in particular, constraints (I) to (IV) are satisfied for the popular CVaR, mean-semideviation, and spectral risk.

It is interesting to compare the performance of the SAA estimator (8) with the analytical-solution based estimator, as in Sections 4.1 and 4.2. In the supplementary material, we report an empirical comparison between the two approaches for the case of CVaR risk, which showed that the two approaches performed very similarly. This is well-expected, since in general, both SAA and standard likelihood-ratio based estimators obey a law-of-large-numbers variance bound of order $1/\sqrt{N}$ [26].

To summarize this section, we have seen that by exploiting the special structure of coherent risk measures in Theorem 2.1 and by the envelope-theorem style result of Theorem 4.2, we are able to derive sampling-based, likelihood-ratio style algorithms for estimating the policy gradient $\nabla_\theta \rho(Z)$ of coherent static risk measures. The gradient estimation algorithms developed here for static risk measures will be used as a sub-routine in our subsequent treatment of dynamic risk measures.

## 5 Gradient Formula for Dynamic Risk

In this section, we derive a new formula for the gradient of the Markov coherent dynamic risk measure, $\nabla_\theta \rho_\infty(\mathcal{M})$. Our approach is based on combining the static gradient formula of Theorem 4.2, with a dynamic-programming decomposition of $\rho_\infty(\mathcal{M})$.

The risk-sensitive *value-function* for an MDP $\mathcal{M}$ under the policy $\theta$ is defined as $V_\theta(x) = \rho_\infty(\mathcal{M}|x_0 = x)$, where with a slight abuse of notation, $\rho_\infty(\mathcal{M}|x_0 = x)$ denotes the Markov-coherent dynamic risk in (3) when the initial state $x_0$ is $x$. It is shown in [24] that due to the structure of the Markov dynamic risk $\rho_\infty(\mathcal{M})$, the value function is the unique solution to the *risk-sensitive Bellman equation*

$$V_\theta(x) = C(x) + \gamma \max_{\xi P_\theta(\cdot|x) \in \mathcal{U}(x, P_\theta(\cdot|x))} \mathbb{E}_\xi[V_\theta(x')], \qquad (9)$$

where the expectation is taken over the next state transition. Note that by definition, we have $\rho_\infty(\mathcal{M}) = V_\theta(x_0)$, and thus, $\nabla_\theta \rho_\infty(\mathcal{M}) = \nabla_\theta V_\theta(x_0)$.

We now develop a formula for $\nabla_\theta V_\theta(x)$; this formula extends the well-known "policy gradient theorem" [28, 14], developed for the expected return, to Markov-coherent dynamic risk measures. We make a standard assumption, analogous to Assumption 4.1 of the static case.

**Assumption 5.1.** *The likelihood ratio $\nabla_\theta \log \mu_\theta(a|x)$ is well-defined and bounded for all $x \in \mathcal{X}$ and $a \in \mathcal{A}$.*

For each state $x \in \mathcal{X}$, let $(\xi_{\theta,x}^*, \lambda_{\theta,x}^{*,\mathcal{P}}, \lambda_{\theta,x}^{*,\mathcal{E}}, \lambda_{\theta,x}^{*,\mathcal{I}})$ denote a saddle point of (6), corresponding to the state $x$, with $P_\theta(\cdot|x)$ replacing $P_\theta$ in (6) and $V_\theta$ replacing $Z$. The next theorem presents a formula for $\nabla_\theta V_\theta(x)$; the proof is in the supplementary material.

**Theorem 5.2.** *Under Assumptions 2.2 and 5.1, we have*

$$\nabla V_\theta(x) = \mathbb{E}_{\xi_\theta^*}\left[\left. \sum_{t=0}^\infty \gamma^t \nabla_\theta \log \mu_\theta(a_t|x_t) h_\theta(x_t, a_t) \right| x_0 = x\right],$$

*where $\mathbb{E}_{\xi_\theta^*}[\cdot]$ denotes the expectation w.r.t. trajectories generated by the Markov chain with transition probabilities $P_\theta(\cdot|x)\xi_{\theta,x}^*(\cdot)$, and the stage-wise cost function $h_\theta(x, a)$ is defined as*

$$h_\theta(x, a) = C(x) + \sum_{x' \in \mathcal{X}} P(x'|x, a)\xi_{\theta,x}^*(x')\left[\gamma V_\theta(x') - \lambda_{\theta,x}^{*,\mathcal{P}} - \sum_{i \in \mathcal{I}} \lambda_{\theta,x}^{*,\mathcal{I}}(i)\frac{df_i(\xi_{\theta,x}^*, p)}{dp(x')} - \sum_{e \in \mathcal{E}} \lambda_{\theta,x}^{*,\mathcal{E}}(e)\frac{dg_e(\xi_{\theta,x}^*, p)}{dp(x')}\right].$$

Theorem 5.2 may be used to develop an *actor-critic* style [28, 14] sampling-based algorithm for solving the DRP problem (5), composed of two interleaved procedures:

**Critic:** For a given policy $\theta$, calculate the risk-sensitive value function $V_\theta$, and
**Actor:** Using the critic's $V_\theta$ and Theorem 5.2, estimate $\nabla_\theta \rho_\infty(\mathcal{M})$ and update $\theta$.

Space limitation restricts us from specifying the full details of our actor-critic algorithm and its analysis. In the following, we highlight only the key ideas and results. For the full details, we refer the reader to the full paper version, provided in the supplementary material.

For the critic, the main challenge is calculating the value function when the state space $\mathcal{X}$ is large and dynamic programming cannot be applied due to the 'curse of dimensionality'. To overcome this, we exploit the fact that $V_\theta$ is equivalent to the value function in a robust MDP [19] and modify a recent algorithm in [31] to estimate it using function approximation.

For the actor, the main challenge is that in order to estimate the gradient using Thm. 5.2, we need to sample from an MDP with $\xi_\theta^*$-weighted transitions. Also, $h_\theta(x, a)$ involves an expectation for each $s$ and $a$. Therefore, we propose a *two-phase sampling procedure* to estimate $\nabla V_\theta$ in which we first use the critic's estimate of $V_\theta$ to derive $\xi_\theta^*$, and sample a trajectory from an MDP with $\xi_\theta^*$-weighted transitions. For each state in the trajectory, we then sample several next states to estimate $h_\theta(x, a)$.

The convergence analysis of the actor-critic algorithm and the gradient error incurred from function approximation of $V_\theta$ are reported in the supplementary material. We remark that our actor-critic algorithm requires a simulator for sampling multiple state-transitions from each state. Extending our approach to work with a single trajectory roll-out is an interesting direction for future research.

## 6 Numerical Illustration

In this section, we illustrate our approach with a numerical example. The purpose of this illustration is to emphasize the importance of *flexibility* in designing risk criteria for selecting an *appropriate* risk-measure – such that suits both the user's risk preference *and* the problem-specific properties.

We consider a trading agent that can invest in one of three assets (see Figure 1 for their distributions). The returns of the first two assets, $A1$ and $A2$, are normally distributed: $A1 \sim \mathcal{N}(1, 1)$ and $A2 \sim$

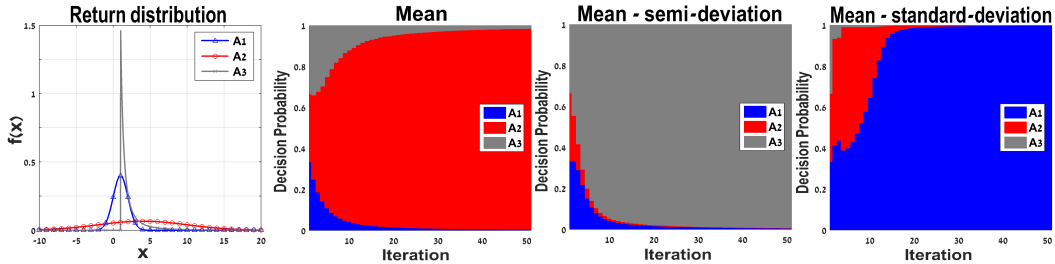

Figure 1: Numerical illustration - selection between 3 assets. A: Probability density of asset return. B,C,D: Bar plots of the probability of selecting each asset vs. training iterations, for policies $\pi_1$, $\pi_2$, and $\pi_3$, respectively. At each iteration, 10,000 samples were used for gradient estimation.

$\mathcal{N}(4, 6)$. The return of the third asset $A3$ has a Pareto distribution: $f(z) = \frac{\alpha}{z^{\alpha+1}} \; \forall z > 1$, with $\alpha = 1.5$. The mean of the return from $A3$ is 3 and its variance is infinite; such heavy-tailed distributions are widely used in financial modeling [22]. The agent selects an action randomly, with probability $P(A_i) \propto \exp(\theta_i)$, where $\theta \in \mathbb{R}^3$ is the policy parameter. We trained three different policies $\pi_1$, $\pi_2$, and $\pi_3$. Policy $\pi_1$ is risk-neutral, i.e., $\max_\theta \mathbb{E}[Z]$, and it was trained using standard policy gradient [15]. Policy $\pi_2$ is risk-averse and had a mean-semideviation objective $\max_\theta \mathbb{E}[Z] - \mathbb{SD}[Z]$, and was trained using the algorithm in Section 4. Policy $\pi_3$ is also risk-averse, with a mean-standard-deviation objective, as proposed in [29, 21], $\max_\theta \mathbb{E}[Z] - \sqrt{\mathrm{Var}[Z]}$, and was trained using the algorithm of [29]. For each of these policies, Figure 1 shows the probability of selecting each asset vs. training iterations. Although $A2$ has the highest mean return, the risk-averse policy $\pi_2$ chooses $A3$, since it has a lower downside, as expected. However, because of the heavy upper-tail of $A3$, policy $\pi_3$ opted to choose $A1$ instead. This is counter-intuitive as a rational investor should not avert high returns. In fact, in this case $A3$ stochastically dominates $A1$ [12].

# 7 Conclusion

We presented algorithms for estimating the gradient of both static and dynamic coherent risk measures using two new policy gradient style formulas that combine sampling with convex programming. Thereby, our approach extends risk-sensitive RL to the whole class of coherent risk measures, and generalizes several recent studies that focused on specific risk measures.

On the technical side, an important future direction is to improve the convergence rate of gradient estimates using importance sampling methods. This is especially important for risk criteria that are sensitive to rare events, such as the CVaR [3].

From a more conceptual point of view, the coherent-risk framework explored in this work provides the decision maker with *flexibility* in designing risk preference. As our numerical example shows, such flexibility is important for selecting appropriate *problem-specific* risk measures for managing the cost variability. However, we believe that our approach has much more potential than that.

In almost every real-world application, uncertainty emanates from stochastic dynamics, but also, and perhaps more importantly, from modeling errors (model uncertainty). A prudent policy should protect against *both* types of uncertainties. The representation duality of coherent-risk (Theorem 2.1), naturally relates the risk to model uncertainty. In [19], a similar connection was made between model-uncertainty in MDPs and dynamic Markov coherent risk. We believe that by carefully shaping the risk-criterion, the decision maker may be able to take uncertainty into account in a *broad* sense. Designing a principled procedure for such *risk-shaping* is not trivial, and is beyond the scope of this paper. However, we believe that there is much potential to risk shaping as it may be the key for handling model misspecification in dynamic decision making.

**Acknowledgments**

The research leading to these results has received funding from the European Research Council under the European Unions Seventh Framework Program (FP7/2007-2013) / ERC Grant Agreement n. 306638. Yinlam Chow is partially supported by Croucher Foundation Doctoral Scholarship.

## Footnotes

[1] Our results may easily be extended to random costs, state-action dependent costs, and random initial states.

[2] For Markov coherent risk, the class of optimal policies is stationary Markov [24], while this is not necessarily true for static risk. Our results can be extended to history-dependent policies or stationary Markov

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
