[Supplementary Material · CoherentRiskNIPS15supp.pdf]

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

[4]In the case when the sizes of state and action spaces are huge or when these spaces are continuous, the empirical transition probability can be found by kernel density estimation.

[5]In the SAA approach, we only sum over the elements for which $P_{\theta;N}(x'|x) > 0$, thus, the sum has at most $N$ elements.

[6]The existence of strict complementary slackness solution follows from the KKT theorem and one can easily construct a strictly complementary pair using i.e. the Balinski-Tucker tableau with the linearized objective function and constraints, in finite time.

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

# A Proof of Theorem 4.2

First note from Assumption 2.2 that

**(i)** Slater's condition holds in the primal optimization problem (1),
**(ii)** $L_\theta(\xi, \lambda^{\mathcal{P}}, \lambda^{\mathcal{E}}, \lambda^{\mathcal{I}})$ is convex in $\xi$ and concave in $(\lambda^{\mathcal{P}}, \lambda^{\mathcal{E}}, \lambda^{\mathcal{I}})$.

Thus by the duality result in convex optimization [9], the above conditions imply strong duality and we have $\rho(Z) = \max_{\xi \geq 0} \min_{\lambda^{\mathcal{P}}, \lambda^{\mathcal{I}} \geq 0, \lambda^{\mathcal{E}}} L_\theta(\xi, \lambda^{\mathcal{P}}, \lambda^{\mathcal{E}}, \lambda^{\mathcal{I}}) = \min_{\lambda^{\mathcal{P}}, \lambda^{\mathcal{I}} \geq 0, \lambda^{\mathcal{E}}} \max_{\xi \geq 0} L_\theta(\xi, \lambda^{\mathcal{P}}, \lambda^{\mathcal{E}}, \lambda^{\mathcal{I}})$. From Assumption 2.2, one can also see that the family of functions $\{L_\theta(\xi, \lambda^{\mathcal{P}}, \lambda^{\mathcal{E}}, \lambda^{\mathcal{I}})\}_{(\xi, \lambda^{\mathcal{P}}, \lambda^{\mathcal{E}}, \lambda^{\mathcal{I}}) \in \mathbb{R}^{|\Omega|} \times \mathbb{R} \times \mathbb{R}^{|\mathcal{E}|} \times \mathbb{R}^{|\mathcal{I}|}}$ is equi-differentiable in $\theta$, $L_\theta(\xi, \lambda^{\mathcal{P}}, \lambda^{\mathcal{E}}, \lambda^{\mathcal{I}})$ is Lipschitz, as a result, an absolutely continuous function in $\theta$, and thus, $\nabla_\theta L_\theta(\xi, \lambda^{\mathcal{P}}, \lambda^{\mathcal{E}}, \lambda^{\mathcal{I}})$ is continuous and bounded at each $(\xi, \lambda^{\mathcal{P}}, \lambda^{\mathcal{E}}, \lambda^{\mathcal{I}})$. Then for every selection of saddle point $(\xi_\theta^*, \lambda_\theta^{*,\mathcal{P}}, \lambda_\theta^{*,\mathcal{E}}, \lambda_\theta^{*,\mathcal{I}}) \in \mathcal{S}$ of (6), using the Envelop theorem for saddle-point problems (see Theorem 4 of [21]), we have

$$\nabla_\theta \max_{\xi \geq 0} \min_{\lambda^{\mathcal{P}}, \lambda^{\mathcal{I}} \geq 0, \lambda^{\mathcal{E}}} L_\theta(\xi, \lambda^{\mathcal{P}}, \lambda^{\mathcal{E}}, \lambda^{\mathcal{I}}) = \nabla_\theta L_\theta(\xi, \lambda^{\mathcal{P}}, \lambda^{\mathcal{E}}, \lambda^{\mathcal{I}})|_{(\xi_\theta^*, \lambda_\theta^{*,\mathcal{P}}, \lambda_\theta^{*,\mathcal{E}}, \lambda_\theta^{*,\mathcal{I}})} . \tag{10}$$

The result follows by writing the gradient in (10) explicitly, and using the likelihood-ratio trick:

$$\sum_{\omega \in \Omega} \xi(\omega) \nabla_\theta P_\theta(\omega) Z(\omega) - \lambda^{\mathcal{P}} \sum_{\omega \in \Omega} \xi(\omega) \nabla_\theta P_\theta(\omega) = \sum_{\omega \in \Omega} \xi(\omega) P(\omega) \nabla_\theta \log P(\omega) \left( Z(\omega) - \lambda^{\mathcal{P}} \right),$$

where the last equality is justified by Assumption 4.1.

# B Gradient Results for Static Mean-Semideviation

In this section we consider the mean-semideviation risk measure, defined as follows:

$$\rho_{\text{MSD}}(Z) = \mathbb{E}[Z] + c \left( \mathbb{E}\left[ (Z - \mathbb{E}[Z])_+^2 \right] \right)^{1/2}, \tag{11}$$

Following the derivation in [32], note that $\left( \mathbb{E}\left[ |Z|^2 \right] \right)^{1/2} = \|Z\|_2$, where $\| \cdot \|_2$ denotes the $L_2$ norm of the space $\mathcal{L}_2(\Omega, \mathcal{F}, P_\theta)$. The norm may also be written as:

$$\|Z\|_2 = \sup_{\|\xi\|_2 \leq 1} \langle \xi, Z \rangle,$$

and hence

$$\left( \mathbb{E}\left[ (Z - \mathbb{E}[Z])_+^2 \right] \right)^{1/2} = \sup_{\|\xi\|_2 \leq 1} \langle \xi, (Z - \mathbb{E}[Z])_+ \rangle = \sup_{\|\xi\|_2 \leq 1, \xi \geq 0} \langle \xi, Z - \mathbb{E}[Z] \rangle$$
$$= \sup_{\|\xi\|_2 \leq 1, \xi \geq 0} \langle \xi - \mathbb{E}[\xi], Z \rangle.$$

It follows that Eq. (1) holds with

$$\mathcal{U} = \{\xi' \in \mathcal{Z}^* : \quad \xi' = 1 + c\xi - c\mathbb{E}[\xi], \quad \|\xi\|_q \leq 1, \quad \xi \geq 0\} .$$

For this case it will be more convenient to write Eq. (1) in the following form

$$\rho_{\text{MSD}}(Z) = \sup_{\|\xi\|_q \leq 1, \xi \geq 0} \langle 1 + c\xi - c\mathbb{E}[\xi], Z \rangle. \tag{12}$$

Let $\bar{\xi}$ denote an optimal solution for (12). In [32] it is shown that $\bar{\xi}$ is a contact point of $(Z - \mathbb{E}[Z])_+$, that is

$$\bar{\xi} \in \arg\max \{ \langle \xi, (Z - \mathbb{E}[Z])_+ \rangle : \|\xi\|_2 \leq 1 \},$$

and we have that

$$\bar{\xi} = \frac{(Z - \mathbb{E}[Z])_+}{\|(Z - \mathbb{E}[Z])_+\|_2} = \frac{(Z - \mathbb{E}[Z])_+}{\mathbb{SD}(Z)}. \tag{13}$$

Note that $\bar{\xi}$ is not necessarily a probability distribution, but for $c \in [0, 1]$, it can be shown [32] that $1 + c\bar{\xi} - c\mathbb{E}[\bar{\xi}]$ always is.

In the following we show that $\bar{\xi}$ may be used to write the gradient $\nabla_\theta \rho_{\text{MSD}}(Z)$ as an expectation, which will lead to a sampling algorithm for the gradient.

**Proposition B.1.** *Under Assumption 4.1, we have that*

$$\nabla_\theta \rho_{MSD}(Z) = \nabla_\theta \mathbb{E}\left[Z\right] + \frac{c}{\mathbb{SD}(Z)}\mathbb{E}\left[(Z - \mathbb{E}\left[Z\right])_+ \left(\nabla_\theta \log P(\omega)(Z - \mathbb{E}\left[Z\right]) - \nabla_\theta \mathbb{E}\left[Z\right]\right)\right],$$

*and, according to the standard likelihood-ratio method,*

$$\nabla_\theta \mathbb{E}\left[Z\right] = \mathbb{E}\left[\nabla_\theta \log P(\omega)Z\right].$$

*Proof.* Note that in Eq. (12) the constraints do not depend on $\theta$. Therefore, using the envelope theorem we obtain that

$$\begin{aligned}
\nabla_\theta \rho(Z) &= \nabla_\theta \langle 1 + c\bar{\xi} - c\mathbb{E}\left[\bar{\xi}\right], Z\rangle \\
&= \nabla_\theta \langle 1, Z\rangle + c\nabla_\theta \langle \bar{\xi}, Z\rangle - c\nabla_\theta \langle \mathbb{E}\left[\bar{\xi}\right], Z\rangle.
\end{aligned} \tag{14}$$

We now write each of the terms in Eq. (14) as an expectation. We start with the following standard likelihood-ratio result:

$$\nabla_\theta \langle 1, Z\rangle = \nabla_\theta \mathbb{E}\left[Z\right] = \mathbb{E}\left[\nabla_\theta \log P(\omega)Z\right].$$

Also, we have that

$$\langle \mathbb{E}\left[\bar{\xi}\right], Z\rangle = \mathbb{E}\left[\bar{\xi}\right]\mathbb{E}\left[Z\right],$$

therefore, by the derivative of a product rule:

$$\nabla_\theta \langle \mathbb{E}\left[\bar{\xi}\right], Z\rangle = \nabla_\theta \mathbb{E}\left[\bar{\xi}\right]\mathbb{E}\left[Z\right] + \mathbb{E}\left[\bar{\xi}\right]\nabla_\theta \mathbb{E}\left[Z\right].$$

By the likelihood-ratio trick and Eq. (13) we have that

$$\nabla_\theta \mathbb{E}\left[\bar{\xi}\right] = \frac{1}{\mathbb{SD}(Z)}\mathbb{E}\left[\nabla_\theta \log P(\omega)(Z - \mathbb{E}\left[Z\right])_+\right].$$

Also, by the likelihood-ratio trick

$$\nabla_\theta \mathbb{E}\left[\bar{\xi}Z\right] = \mathbb{E}\left[\nabla_\theta \log P(\omega)\bar{\xi}Z\right].$$

Plugging these terms back in Eq. (14), we have that

$$\begin{aligned}
\nabla_\theta \rho(Z) &= \nabla_\theta \mathbb{E}\left[Z\right] + c\nabla_\theta \mathbb{E}\left[\bar{\xi}Z\right] - c\nabla_\theta \mathbb{E}\left[\bar{\xi}\right]\mathbb{E}\left[Z\right] - c\mathbb{E}\left[\bar{\xi}\right]\nabla_\theta \mathbb{E}\left[Z\right] \\
&= \nabla_\theta \mathbb{E}\left[Z\right] + c\mathbb{E}\left[\bar{\xi}\left(\nabla_\theta \log P(\omega)Z - \nabla_\theta \mathbb{E}\left[Z\right]\right)\right] - c\nabla_\theta \mathbb{E}\left[\bar{\xi}\right]\mathbb{E}\left[Z\right] \\
&= \nabla_\theta \mathbb{E}\left[Z\right] + \frac{c}{\mathbb{SD}(Z)}\mathbb{E}\left[(Z - \mathbb{E}\left[Z\right])_+ \left(\nabla_\theta \log P(\omega)Z - \nabla_\theta \mathbb{E}\left[Z\right]\right)\right] - c\nabla_\theta \mathbb{E}\left[\bar{\xi}\right]\mathbb{E}\left[Z\right] \\
&= \nabla_\theta \mathbb{E}\left[Z\right] + \frac{c}{\mathbb{SD}(Z)}\mathbb{E}\left[(Z - \mathbb{E}\left[Z\right])_+ \left(\nabla_\theta \log P(\omega)(Z - \mathbb{E}\left[Z\right]) - \nabla_\theta \mathbb{E}\left[Z\right]\right)\right].
\end{aligned}$$

□

Proposition 4.3 naturally leads to a sampling-based gradient estimation algortihm, which we term `GMSD` (Gradient of Mean Semi-Deviation). The algorithm is described in Algorithm 1.

## C   Consistency Proof

Let $(\Omega_{SAA}, \mathcal{F}_{SAA}, P_{SAA})$ denote the probability space of the SAA functions (i.e., the randomness due to sampling).

Let $L_{\theta;N}(\xi, \lambda^{\mathcal{P}}, \lambda^{\mathcal{E}}, \lambda^{\mathcal{I}})$ denote the Lagrangian of the SAA problem

$$\begin{aligned}
L_{\theta;N}(\xi, \lambda^{\mathcal{P}}, \lambda^{\mathcal{E}}, \lambda^{\mathcal{I}}) = \sum_{\omega \in \Omega}\xi(\omega)P_{\theta;N}(\omega)Z(\omega) - \lambda^{\mathcal{P}}\left(\sum_{\omega \in \Omega}\xi(\omega)P_{\theta;N}(\omega) - 1\right) \\
- \sum_{e \in \mathcal{E}}\lambda^{\mathcal{E}}(e)f_e(\xi, P_{\theta;N}) - \sum_{i \in \mathcal{I}}\lambda^{\mathcal{I}}(i)f_i(\xi, P_{\theta;N}).
\end{aligned} \tag{15}$$

Recall that $\mathcal{S} \subset \mathbb{R}^{|\Omega|} \times \mathbb{R} \times \mathbb{R}^{|\mathcal{E}|} \times \mathbb{R}_+^{|\mathcal{I}|}$ denotes the set of saddle points of the true Lagrangian (6). Let $\mathcal{S}_N \subset \mathbb{R}^{|\Omega|} \times \mathbb{R} \times \mathbb{R}^{|\mathcal{E}|} \times \mathbb{R}_+^{|\mathcal{I}|}$ denote the set of SAA Lagrangian (15) saddle points.

Suppose that there exists a compact set $C \equiv C_\xi \times C_\lambda$, where $C_\xi \subset \mathbb{R}^{|\Omega|}$ and $C_\lambda \subset \mathbb{R} \times \mathbb{R}^{|\mathcal{E}|} \times \mathbb{R}_+^{|\mathcal{I}|}$ such that:

---

**Algorithm 1** GMSD

---

1: **Given:**

      • Risk level $c$

      • An i.i.d. sequence $z_1, \ldots, z_N \sim P_\theta$.

2: Set

$$\widehat{\mathbb{E}\left[Z\right]} = \frac{1}{N} \sum_{i=1}^{N} z_i.$$

3: Set

$$\widehat{\mathbb{SD}(Z)} = \left( \frac{1}{N} \sum_{i=1}^{N} (z_i - \widehat{\mathbb{E}\left[Z\right]})_+^2 \right)^{1/2}.$$

4: Set

$$\widehat{\nabla_\theta \mathbb{E}\left[Z\right]} = \frac{1}{N} \sum_{i=1}^{N} \nabla_\theta \log P(z_i) z_i.$$

5: **Return:**

$$\nabla_\theta \hat\rho(Z) = \widehat{\nabla_\theta \mathbb{E}\left[Z\right]} + \frac{c}{\widehat{\mathbb{SD}(Z)}} \frac{1}{N} \sum_{i=1}^{N} (z_i - \widehat{\mathbb{E}\left[Z\right]})_+ \left( \nabla_\theta \log P(z_i)(z_i - \widehat{\mathbb{E}\left[Z\right]}) - \widehat{\nabla_\theta \mathbb{E}\left[Z\right]} \right).$$

---

**(i)** The set of Lagrangian saddle points $\mathcal{S} \subset C$ is non-empty and bounded.

**(ii)** The functions $f_e(\xi, P_\theta)$ for all $e \in \mathcal{E}$ and $f_i(\xi, P_\theta)$ for all $i \in \mathcal{I}$ are finite valued and continuous (in $\xi$) on $C_\xi$.

**(iii)** For $N$ large enough the set $\mathcal{S}_N$ is non-empty and $\mathcal{S}_N \subset C$ w.p. 1.

Recall from Assumption 2.2 that for each fixed $\xi \in \mathcal{B}$, both $f_i(\xi, p)$ and $g_e(\xi, p)$ are continuous in $p$. Furthermore, by the S.L.L.N. of Markov chains, for each policy parameter, we have $P_{\theta,N} \to P_\theta$ w.p. 1. From the definition of the Lagrangian function and continuity of constraint functions, one can easily see that for each $(\xi, \lambda^\mathcal{P}, \lambda^\mathcal{E}, \lambda^\mathcal{I}) \in \mathbb{R}^{|\Omega|} \times \mathbb{R} \times \mathbb{R}^{|\mathcal{E}|} \times \mathbb{R}_+^{|\mathcal{I}|}$, $L_{\theta;N}(\xi, \lambda^\mathcal{P}, \lambda^\mathcal{E}, \lambda^\mathcal{I}) \to L_\theta(\xi, \lambda^\mathcal{P}, \lambda^\mathcal{E}, \lambda^\mathcal{I})$ w.p. 1. Denote with $\mathbb{D}\{A, B\}$ the deviation of set $A$ from set $B$, i.e., $\mathbb{D}\{A, B\} = \sup_{x \in A} \inf_{y \in B} \|x - y\|$. Further assume that:

**(iv)** If $\xi_N \in \mathcal{U}(P_{\theta;N})$ and $\xi_N$ converges w.p. 1 to a point $\xi$, then $\xi \in \mathcal{U}(P_\theta)$.

According to the discussion in Page 161 of [32], the Slater condition of Assumption 2.2 guarantees the following condition:

**(v)** For some point $\xi \in \mathcal{P}$ there exists a sequence $\xi_N \in \mathcal{U}(P_{\theta;N})$ such that $\xi_N \to \xi$ w.p. 1,

and from Theorem 6.6 in [32], we know that both sets $\mathcal{U}(P_{\theta;N})$ and $\mathcal{U}(P_\theta)$ are convex and compact. Furthermore, note that we have

**(vi)** The objective function on (1) is linear, finite valued and continuous in $\xi$ on $C_\xi$ (these conditions obviously hold for almost all $\omega \in \Omega$ in the integrand function $\xi(\omega)Z(\omega)$).

**(vii)** S.L.L.N. holds point-wise for any $\xi$.

From (i,iv,v,vi,vii), and under the same lines of proof as in Theorem 5.5 of [32], we have that

$$\rho_N(Z) \to \rho(Z) \text{ w.p. 1 as } N \to \infty, \tag{16}$$

$$\mathbb{D}\{\mathcal{P}_N, \mathcal{P}\} \to 0 \text{ w.p. 1 as } N \to \infty, \tag{17}$$

In part 1 and part 2 of the following proof, we show, by following similar derivations as in Theorem 5.2, Theorem 5.3 and Theorem 5.4 of [32], that $L_{\theta;N}(\xi^*_{\theta;N}, \lambda^{*,\mathcal{P}}_{\theta;N}, \lambda^{*,\mathcal{E}}_{\theta;N}, \lambda^{*,\mathcal{I}}_{\theta;N}) \to$

$L_\theta(\xi_\theta^*, \lambda_\theta^{*,\mathcal{P}}, \lambda_\theta^{*,\mathcal{E}}, \lambda_\theta^{*,\mathcal{I}})$ w.p. 1 and $\mathbb{D}\{\mathcal{S}_N, \mathcal{S}\} \to 0$ w.p. 1 as $N \to \infty$. Based on the definition of the deviation of sets, the limit point of any element in $\mathcal{S}_N$ is also an element in $\mathcal{S}$.

Assumptions (i) and (iii) imply that we can restrict our attention to the set $C$.

**Part 1** We first show that $L_{\theta;N}(\xi_{\theta;N}^*, \lambda_{\theta;N}^{*,\mathcal{P}}, \lambda_{\theta;N}^{*,\mathcal{E}}, \lambda_{\theta;N}^{*,\mathcal{I}})$ converges to $L_\theta(\xi_\theta^*, \lambda_\theta^{*,\mathcal{P}}, \lambda_\theta^{*,\mathcal{E}}, \lambda_\theta^{*,\mathcal{I}})$ w.p. 1 as $N \to \infty$.

For each fixed $(\lambda^{\mathcal{P}}, \lambda^{\mathcal{E}}, \lambda^{\mathcal{I}}) \in C_\lambda$, the function $L_\theta(\xi, \lambda^{\mathcal{P}}, \lambda^{\mathcal{E}}, \lambda^{\mathcal{I}})$ is convex and continuous in $\xi$. Together with the point-wise S.L.L.N. property, Theorem 7.49 of [32] implies that $L_{\theta;N}(\xi, \lambda^{\mathcal{P}}, \lambda^{\mathcal{E}}, \lambda^{\mathcal{I}}) - L_\theta(\xi, \lambda^{\mathcal{P}}, \lambda^{\mathcal{E}}, \lambda^{\mathcal{I}}) \xrightarrow{e} 0$, where $\xrightarrow{e}$ denotes epi-convergence. Furthermore, since the objective and constraint functions are convex in $\xi$ and are finite valued on $C_\xi$, the set $\text{dom} L_\theta(\cdot, \lambda^{\mathcal{P}}, \lambda^{\mathcal{E}}, \lambda^{\mathcal{I}})$ has non-empty interior. It follows from Theorem 7.27 of [32] that epi-convergence of $L_{\theta,N}$ to $L_\theta$ implies uniform convergence on $C_\xi$, i.e., $\sup_{\xi \in C_\xi} |L_{\theta;N}(\xi, \lambda^{\mathcal{P}}, \lambda^{\mathcal{E}}, \lambda^{\mathcal{I}}) - L_\theta(\xi, \lambda^{\mathcal{P}}, \lambda^{\mathcal{E}}, \lambda^{\mathcal{I}})| \leq \epsilon$. On the other hand, for each fixed $\xi \in C_\xi$, the function $L_\theta(\xi, \lambda^{\mathcal{P}}, \lambda^{\mathcal{E}}, \lambda^{\mathcal{I}})$ is linear and thus continuous in $(\lambda^{\mathcal{P}}, \lambda^{\mathcal{E}}, \lambda^{\mathcal{I}})$ and $\text{dom} L_\theta(\xi, \cdot, \cdot, \cdot) = \mathbb{R} \times \mathbb{R}^{|\mathcal{E}|} \times \mathbb{R}^{|\mathcal{I}|}$ has non-empty interior. It follows from analogous arguments that $\sup_{(\lambda^{\mathcal{P}}, \lambda^{\mathcal{E}}, \lambda^{\mathcal{I}}) \in C_\lambda} |L_{\theta;N}(\xi, \lambda^{\mathcal{P}}, \lambda^{\mathcal{E}}, \lambda^{\mathcal{I}}) - L_\theta(\xi, \lambda^{\mathcal{P}}, \lambda^{\mathcal{E}}, \lambda^{\mathcal{I}})| \leq \epsilon$. Combining these results implies that for any $\epsilon > 0$ and a.e. $\omega_{SAA} \in \Omega_{SAA}$ there is a $N^*(\epsilon, \omega_{SAA})$ such that

$$\sup_{(\xi, \lambda^{\mathcal{P}}, \lambda^{\mathcal{E}}, \lambda^{\mathcal{I}}) \in C} |L_{\theta;N}(\xi, \lambda^{\mathcal{P}}, \lambda^{\mathcal{E}}, \lambda^{\mathcal{I}}) - L_\theta(\xi, \lambda^{\mathcal{P}}, \lambda^{\mathcal{E}}, \lambda^{\mathcal{I}})| \leq \epsilon. \tag{18}$$

Now, assume by contradiction that for some $N > N^*(\epsilon, \omega_{SAA})$ we have $L_{\theta;N}(\xi_{\theta;N}^*, \lambda_{\theta;N}^{*,\mathcal{P}}, \lambda_{\theta;N}^{*,\mathcal{E}}, \lambda_{\theta;N}^{*,\mathcal{I}}) - L_\theta(\xi_\theta^*, \lambda_\theta^{*,\mathcal{P}}, \lambda_\theta^{*,\mathcal{E}}, \lambda_\theta^{*,\mathcal{I}}) > \epsilon$. Then by definition of the saddle points

$$L_{\theta;N}(\xi_{\theta;N}^*, \lambda_\theta^{*,\mathcal{P}}, \lambda_\theta^{*,\mathcal{E}}, \lambda_\theta^{*,\mathcal{I}}) \geq L_{\theta;N}(\xi_{\theta;N}^*, \lambda_{\theta;N}^{*,\mathcal{P}}, \lambda_{\theta;N}^{*,\mathcal{E}}, \lambda_{\theta;N}^{*,\mathcal{I}})$$
$$> L_\theta(\xi_\theta^*, \lambda_\theta^{*,\mathcal{P}}, \lambda_\theta^{*,\mathcal{E}}, \lambda_\theta^{*,\mathcal{I}}) + \epsilon \geq L_\theta(\xi_{\theta;N}^*, \lambda_\theta^{*,\mathcal{P}}, \lambda_\theta^{*,\mathcal{E}}, \lambda_\theta^{*,\mathcal{I}}) + \epsilon,$$

contradicting (18).

Similarly, assuming by contradiction that $L_\theta(\xi_\theta^*, \lambda_\theta^{*,\mathcal{P}}, \lambda_\theta^{*,\mathcal{E}}, \lambda_\theta^{*,\mathcal{I}}) - L_{\theta;N}(\xi_{\theta;N}^*, \lambda_{\theta;N}^{*,\mathcal{P}}, \lambda_{\theta;N}^{*,\mathcal{E}}, \lambda_{\theta;N}^{*,\mathcal{I}}) > \epsilon$ gives

$$L_\theta(\xi_\theta^*, \lambda_{\theta;N}^{*,\mathcal{P}}, \lambda_{\theta;N}^{*,\mathcal{E}}, \lambda_{\theta;N}^{*,\mathcal{I}}) \geq L_\theta(\xi_\theta^*, \lambda_\theta^{*,\mathcal{P}}, \lambda_\theta^{*,\mathcal{E}}, \lambda_\theta^{*,\mathcal{I}})$$
$$> L_{\theta;N}(\xi_{\theta;N}^*, \lambda_{\theta;N}^{*,\mathcal{P}}, \lambda_{\theta;N}^{*,\mathcal{E}}, \lambda_{\theta;N}^{*,\mathcal{I}}) + \epsilon \geq L_{\theta;N}(\xi_\theta^*, \lambda_{\theta;N}^{*,\mathcal{P}}, \lambda_{\theta;N}^{*,\mathcal{E}}, \lambda_{\theta;N}^{*,\mathcal{I}}) + \epsilon,$$

also contradicting (18).

It follows that $\left| L_{\theta;N}(\xi_{\theta;N}^*, \lambda_{\theta;N}^{*,\mathcal{P}}, \lambda_{\theta;N}^{*,\mathcal{E}}, \lambda_{\theta;N}^{*,\mathcal{I}}) - L_\theta(\xi_\theta^*, \lambda_\theta^{*,\mathcal{P}}, \lambda_\theta^{*,\mathcal{E}}, \lambda_\theta^{*,\mathcal{I}}) \right| \leq \epsilon$ for all $N > N^*(\epsilon, \omega_{SAA})$, and therefore

$$\lim_{N \to \infty} L_{\theta;N}(\xi_{\theta;N}^*, \lambda_{\theta;N}^{*,\mathcal{P}}, \lambda_{\theta;N}^{*,\mathcal{E}}, \lambda_{\theta;N}^{*,\mathcal{I}}) = L_\theta(\xi_\theta^*, \lambda_\theta^{*,\mathcal{P}}, \lambda_\theta^{*,\mathcal{E}}, \lambda_\theta^{*,\mathcal{I}}), \tag{19}$$

w.p. 1.

**Part 2** Let us now show that $\mathbb{D}\{\mathcal{S}_N, \mathcal{S}\} \to 0$. We argue by a contradiction. Suppose that $\mathbb{D}\{\mathcal{S}_N, \mathcal{S}\} \not\to 0$. Since $C$ is compact, we can assume that there exists a sequence $(\xi_{\theta;N}^*, \lambda_{\theta;N}^{*,\mathcal{P}}, \lambda_{\theta;N}^{*,\mathcal{E}}, \lambda_{\theta;N}^{*,\mathcal{I}}) \in \mathcal{S}_N$ that converges to a point $(\bar{\xi}^*, \bar{\lambda}^{*,\mathcal{P}}, \bar{\lambda}^{*,\mathcal{E}}, \bar{\lambda}^{*,\mathcal{I}}) \in C$ and $(\bar{\xi}^*, \bar{\lambda}^{*,\mathcal{P}}, \bar{\lambda}^{*,\mathcal{E}}, \bar{\lambda}^{*,\mathcal{I}}) \notin \mathcal{S}$. However, from (17) we must have that $\bar{\xi}^* \in \mathcal{P}$. Therefore, we must have that

$$L_\theta(\bar{\xi}^*, \bar{\lambda}^{*,\mathcal{P}}, \bar{\lambda}^{*,\mathcal{E}}, \bar{\lambda}^{*,\mathcal{I}}) > L_\theta(\bar{\xi}^*, \lambda_\theta^{*,\mathcal{P}}, \lambda_\theta^{*,\mathcal{E}}, \lambda_\theta^{*,\mathcal{I}}),$$

by definition of the saddle point set.

Now,

$$L_{\theta;N}(\xi_{\theta;N}^*, \lambda_{\theta;N}^{*,\mathcal{P}}, \lambda_{\theta;N}^{*,\mathcal{E}}, \lambda_{\theta;N}^{*,\mathcal{I}}) - L_\theta(\bar{\xi}^*, \bar{\lambda}^{*,\mathcal{P}}, \bar{\lambda}^{*,\mathcal{E}}, \bar{\lambda}^{*,\mathcal{I}})$$
$$= \left[ L_{\theta;N}(\xi_{\theta;N}^*, \lambda_{\theta;N}^{*,\mathcal{P}}, \lambda_{\theta;N}^{*,\mathcal{E}}, \lambda_{\theta;N}^{*,\mathcal{I}}) - L_\theta(\xi_{\theta;N}^*, \lambda_{\theta;N}^{*,\mathcal{P}}, \lambda_{\theta;N}^{*,\mathcal{E}}, \lambda_{\theta;N}^{*,\mathcal{I}}) \right] +$$
$$+ \left[ L_\theta(\xi_{\theta;N}^*, \lambda_{\theta;N}^{*,\mathcal{P}}, \lambda_{\theta;N}^{*,\mathcal{E}}, \lambda_{\theta;N}^{*,\mathcal{I}}) - L_\theta(\bar{\xi}^*, \bar{\lambda}^{*,\mathcal{P}}, \bar{\lambda}^{*,\mathcal{E}}, \bar{\lambda}^{*,\mathcal{I}}) \right]. \tag{20}$$

The first term in the r.h.s. of (20) tends to zero, using the argument from (18), and the second by continuity of $L_\theta$ guaranteed by (ii). We thus obtain that $L_{\theta;N}(\xi^*_{\theta;N}, \lambda^{*,\mathcal{P}}_{\theta;N}, \lambda^{*,\mathcal{E}}_{\theta;N}, \lambda^{*,\mathcal{I}}_{\theta;N})$ tends to $L_\theta(\bar{\xi}^*, \bar{\lambda}^{*,\mathcal{P}}, \bar{\lambda}^{*,\mathcal{E}}, \bar{\lambda}^{*,\mathcal{I}}) > L_\theta(\xi^*_\theta, \lambda^{*,\mathcal{P}}_\theta, \lambda^{*,\mathcal{E}}_\theta, \lambda^{*,\mathcal{I}}_\theta)$, which is a contradiction to (19).

**Part 3** We now show the consistency of $\nabla_{\theta;N}\rho(Z)$.

Consider Eq. (8). Since $\nabla_\theta \log P(\cdot)$ is bounded by Assumption 4.1, and $\nabla_\theta f_i(\cdot; P_\theta)$ and $\nabla_\theta g_e(\cdot; P_\theta)$ are bounded by Assumption 2.2, and using our previous result $\mathbb{D}\{\mathcal{S}_N, \mathcal{S}\} \to 0$, we have that for a.e. $\omega_{SAA} \in \Omega_{SAA}$

$$\begin{aligned}
\lim_{N\to\infty} \nabla_{\theta;N}\rho(Z) &= \sum_{\omega\in\Omega} P_\theta(\omega)\xi^*_\theta(\omega)\nabla_\theta \log P(\omega)(Z(\omega) - \lambda^{*,\mathcal{P}}_\theta) \\
&\quad - \sum_{e\in\mathcal{E}} \lambda^{*,\mathcal{E}}_\theta(e)\nabla_\theta g_e(\xi^*_\theta; P_\theta) \\
&\quad - \sum_{i\in\mathcal{I}} \lambda^{*,\mathcal{I}}_\theta(i)\nabla_\theta f_i(\xi^*_\theta; P_\theta) \\
&= \nabla_\theta \rho(Z).
\end{aligned}$$

where the first equality is obtained from the Envelop theorem (see Theorem 4.2) with $(\xi^*_\theta, \lambda^{*,\mathcal{P}}_\theta, \lambda^{*,\mathcal{E}}_\theta, \lambda^{*,\mathcal{I}}_\theta) \in \mathcal{S}_N \cap \mathcal{S}$ is the limit point of the converging sequence $\{(\xi^*_{\theta;N}, \lambda^{*,\mathcal{P}}_{\theta;N}, \lambda^{*,\mathcal{E}}_{\theta;N}, \lambda^{*,\mathcal{I}}_{\theta;N})\}_{N\in\mathbb{N}}$.

# D  Empirical Comparison of Analytic-Solution Based and SAA Based Policy Gradient

We compare the CVaR policy gradient as obtained by the analytical result in Section 4.1:

$$\nabla_\theta \rho_{\text{CVaR}}(Z; \alpha) = \mathbb{E}\left[\nabla_\theta \log P(\omega)(Z - q_\alpha)\,|\, Z(\omega) > q_\alpha\right], \tag{21}$$

with the general sampling based algorithm of Eq. (8) in Section 4.3.

For the analytical-solution based policy gradient, we use the GCVaR algorithm of [30], which is the sampling-based version of Eq. (21). For the general sampling based algorithm, we used Matlab's 'linprog' to solve the linear program in Eq. (7), using the risk envelope for CVaR, as defined in Section 4.1. The resulting numerical values for $\xi^*_{\theta;N}$ and $\lambda^{*,\mathcal{P}}_{\theta;N}$ were plugged into Eq. (8) for the gradient estimate (the other terms in Eq. (8) cancel out by definition of the CVaR risk envelope).

We present empirical results for the asset selection domain of Section 6. We chose a CVaR level of $\alpha = 0.05$ (corresponding to the average of the worst 5% outcomes), and trained policies with either the analytical-solution based policy gradient (labeled CVaR), and the general sampling based algorithm (labeled CVaRS). In Figure 2 we plot the learning curves (the $\theta$ values vs. training episodes) of both policies, for different values of $N$ - the sampling budget.

As may be observed, both policies exhibit similar learning performance, and the differences diminish as $N$ grows large.

Figure 2: Learning curves ($\theta$ values vs. training episodes) of the analytical-solution based policy gradient (labeled CVaR), and the general sampling based algorithm (labeled CVaRS), for different values of $N$ - the sampling budget.

## E    Proof of Theorem 5.2

Similar to the proof of Theorem 4.2, recall the saddle point definition of $(\xi_{\theta,x}^{*}, \lambda_{\theta,x}^{*,\mathcal{P}}, \lambda_{\theta,x}^{*,\mathcal{E}}, \lambda_{\theta,x}^{*,\mathcal{I}}) \in \mathcal{S}$ and strong duality result, i.e.,

$$\max_{\xi \, : \, \xi P_\theta(\cdot|x) \in \mathcal{U}(x, P_\theta(\cdot|x))} \sum_{x' \in \mathcal{X}} \xi(x') P_\theta(x'|x) V_\theta(x') = \max_{\xi \geq 0} \min_{\lambda^\mathcal{P}, \lambda^\mathcal{I} \geq 0, \lambda^\mathcal{E}} L_{\theta,x}(\xi, \lambda^\mathcal{P}, \lambda^\mathcal{E}, \lambda^\mathcal{I})$$

$$= \min_{\lambda^\mathcal{P}, \lambda^\mathcal{I} \geq 0, \lambda^\mathcal{E}} \max_{\xi \geq 0} L_{\theta,x}(\xi, \lambda^\mathcal{P}, \lambda^\mathcal{E}, \lambda^\mathcal{I}).$$

the gradient formula in (10) can be written as

$$\nabla_\theta V_\theta(x) = \nabla_\theta \left[ C_\theta(x) + \gamma \max_{\xi \, : \, \xi P_\theta(\cdot|x) \in \mathcal{U}(x, P_\theta(\cdot|x))} \mathbb{E}_\xi[V_\theta] \right]$$

$$= \gamma \sum_{x' \in \mathcal{X}} \xi_{\theta,x}^{*}(x') P_\theta(x'|x) \nabla_\theta V_\theta(x') + \sum_{a \in \mathcal{A}} \mu_\theta(a|x) \nabla_\theta \log \mu_\theta(a|x) h_\theta(x,a),$$

where the stage-wise cost function $h_\theta(x,a)$ is defined in (27).   By defining $\widehat{h}_\theta(x) = \sum_{a \in \mathcal{A}} \mu_\theta(a|x) \nabla_\theta \log \mu_\theta(a|x) h_\theta(x,a)$ and unfolding the recursion, the above expression implies

$$\nabla_\theta V_\theta(x_0) = \widehat{h}_\theta(x_0) + \gamma \sum_{x_1 \in \mathcal{X}} P_\theta(x_1|x_0) \xi_\theta^{*}(x_1) \left[ \widehat{h}_\theta(x_1) + \gamma \sum_{x_2 \in \mathcal{X}} P_\theta(x_2|x_1) \xi_\theta^{*}(x_2) \nabla_\theta V_\theta(x_2) \right].$$

Now since $\nabla_\theta V_\theta$ is continuously differentiable with bounded derivatives, when $t \to \infty$, one obtains $\gamma^t \nabla_\theta V_\theta(x) \to 0$ for any $x \in \mathcal{X}$. Therefore, by Bounded Convergence Theorem, $\lim_{t \to \infty} \rho(\gamma^t V_\theta(x_t)) = 0$, when $x_0 = x$ the above expression implies the result of this theorem.

# F   Gradient Formula for Dynamic Risk - Full Results

In this section, we first derive a new formula for the gradient of a general Markov-coherent dynamic risk measure $\nabla_\theta \rho_\infty(\mathcal{M})$ that involves the *value function* of the risk objective $\rho_\infty(\mathcal{M})$ (e.g., the value function proposed by [30]). This formula extends the well-known "policy gradient theorem" [34, 17] developed for the expected return to Markov-coherent dynamic risk measures. Using this formula, we suggest the following actor-critic style algorithm for estimating $\nabla_\theta \rho_\infty(\mathcal{M})$:

**Critic:** For a given policy $\theta$, calculate the *risk-sensitive value function* of $\rho_\infty(\mathcal{M})$ (see Section F.3), and
**Actor:** Using the critic's value function, estimate $\nabla_\theta \rho_\infty(\mathcal{M})$ by sampling (see Section F.4).

The value function proposed by [30] assigns to each state a particular value that encodes the long-term risk starting from that state. When the state space $\mathcal{X}$ is large, calculating the value function by dynamic programming (as suggested by [30]) becomes intractable due to the "curse of dimensionality". For the risk-neutral case, a standard solution to this problem is to approximate the value function by a set of state-dependent features, and use sampling to calculate the parameters of this approximation [6]. In particular, *temporal difference* (TD) learning methods [33] are popular for this purpose, which have been recently extended to robust MDPs by [37]. We use their (robust) TD algorithm and show how our critic use it to approximates the *risk-sensitive* value function. We then discuss how the error introduced by this approximation affects the gradient estimate of the actor.

## F.1   Dynamic Risk

We provide a multi-period generalization of the concepts presented in Section 2.1. Here we closely follow the discussion in [30].

Consider a probability space $(\Omega, \mathcal{F}, P_\theta)$, a filtration $\mathcal{F}_0 \subset \mathcal{F}_1 \subset \mathcal{F}_2 \cdots \subset \mathcal{F}_T \subset \mathcal{F}$, and an adapted sequence of real-valued random variables $Z_t$, $t \in \{0, \dots, T\}$. We assume that $\mathcal{F}_0 = \{\Omega, \emptyset\}$, i.e., $Z_0$ is deterministic. For each $t \in \{0, \dots, T\}$, we denote by $\mathcal{Z}_t$ the space of random variables defined over the probability space $(\Omega, \mathcal{F}_t, P_\theta)$, and also let $\mathcal{Z}_{t,T} := \mathcal{Z}_t \times \cdots \times \mathcal{Z}_T$ be a sequence of these spaces. The sequence of random variables $Z_t$ can be interpreted as the stage-wise costs observed along a trajectory generated by an MDP parameterized by a parameter $\theta$, i.e., $Z_{0,T} \doteq \big(Z_0 = \gamma^0 C(x_0, a_0), \dots, Z_T = \gamma^T C(x_T, a_T)\big) \in \mathcal{Z}_{0,T}$.

In particular, we are interested in the sequence of random variables induced by the trajectories from a Markov decision process (MDP) parameterized by parameter $\theta$.

Explicitly, for any $t \geq 0$ and state dependent random variable $Z(x_{t+1}) \in \mathcal{Z}_{t+1}$, the risk evaluation is given by

$$\rho\big(Z(x_{t+1})\big) = \max_{\xi \,:\, \xi P_\theta(\cdot|x_t) \in \mathcal{U}(x_t, P_\theta(\cdot|x_t))} \mathbb{E}_\xi\big[Z(x_{t+1})\big], \tag{22}$$

where we let $\mathcal{U}(x_t, P_\theta(\cdot|x_t))$ denote the risk-envelope (2) with $P_\theta$ replaced with $P_\theta(\cdot|x_t)$. The Markovian assumption on the risk measure $\rho_T(\mathcal{M})$ allows us to optimize it using dynamic programming techniques.

## F.2   Risk-Sensitive Bellman Equation

Our value-function estimation method is driven by a Bellman-style equation for Markov coherent risks. Let $B(\mathcal{X})$ denote the space of real-valued bounded functions on $\mathcal{X}$ and $C_\theta(x) = \sum_{a \in \mathcal{A}} C(x,a)\mu_\theta(a|x)$ be the stage-wise cost function induced by policy $\mu_\theta$. We now define the risk sensitive Bellman operator $T_\theta[V] : B(\mathcal{X}) \mapsto B(\mathcal{X})$ as

$$T_\theta[V](x) := C_\theta(x) + \gamma \max_{\xi P_\theta(\cdot|x) \in \mathcal{U}(x, P_\theta(\cdot|x))} \mathbb{E}_\xi[V]. \tag{23}$$

According to Theorem 1 in [30], the operator $T_\theta$ has a unique fixed-point $V_\theta$, i.e., $T_\theta[V_\theta](x) = V_\theta(x)$, $\forall x \in \mathcal{X}$, that is equal to the risk objective function induced by $\theta$, i.e., $V_\theta(x_0) = \rho_\infty(\mathcal{M})$. However, when the state space $\mathcal{X}$ is large, exact enumeration of the Bellman equation is intractable

due to "curse of dimensionality". Next, we provide an iterative approach to approximate the risk sensitive value function.

## F.3 Value Function Approximation

Consider the linear approximation of the risk-sensitive value function $V_\theta(x) \approx v^\top \phi(x)$, where $\phi(\cdot) \in \mathbb{R}^{\kappa_2}$ is the $\kappa_2$-dimensional state-dependent feature vector. Thus, the approximate value function belongs to the low dimensional sub-space $\mathcal{V} = \{\Phi v | v \in \mathbb{R}^{\kappa_2}\}$, where $\Phi : \mathcal{X} \to \mathbb{R}^{\kappa_2}$ is a function mapping such that $\Phi(x) = \phi(x)$. The goal of our critic is to find a good approximation of $V_\theta$ from simulated trajectories of the MDP. In order to have a well-defined approximation scheme, we first impose the following standard assumption [6].

**Assumption F.1.** *The mapping $\Phi$ has full column rank.*

For a function $y : \mathcal{X} \to \mathbb{R}$, we define its weighted (by $d$) $\ell_2$-norm as $\|y\|_d = \sqrt{\sum_{x'} d(x'|x)y(x')^2}$, where $d$ is a distribution over $\mathcal{X}$. Using this, we define $\Pi : \mathcal{X} \to \mathcal{V}$, the orthogonal projection from $\mathbb{R}$ to $\mathcal{V}$, w.r.t. a norm weighted by the stationary distribution of the policy, $d_\theta(x'|x)$.

Note that the TD methods approximate the value function $V_\theta$ with the fixed-point of the joint operator $\Pi T_\theta$, i.e., $\tilde{V}_\theta(x) = v_\theta^{*\top}\phi(x)$, such that

$$\forall x \in \mathcal{X}, \qquad \tilde{V}_\theta(x) = \Pi T_\theta[\tilde{V}_\theta](x). \tag{24}$$

From Eq. 22 that has been derived from Theorem 2.1 for dynamic risks, it is easy to see that the risk-sensitive Bellman equation (23) is a robust Bellman equation [23] with uncertainty set $\mathcal{U}(x, P_\theta(\cdot|x))$. Thus, we may use the TD approximation of the robust Bellman equation proposed by [37] to find an approximation of $V_\theta$. We will need the following assumption analogous to Assumption 2 in [37].

**Assumption F.2.** *There exists $\kappa \in (0,1)$ such that $\xi(x') \leq \kappa/\gamma$, for all $\xi(\cdot)P_\theta(\cdot|x) \in \mathcal{U}(x, P_\theta(\cdot|x))$ and all $x, x' \in \mathcal{X}$.*

Given Assumption F.2, Proposition 3 in [37] guarantees that the projected risk-sensitive Bellman operator $\Pi T_\theta$ is a contraction w.r.t. $d_\theta$-norm. Therefore, Eq. 24 has a unique fixed-point solution $\tilde{V}_\theta(x) = v_\theta^{*\top}\phi(x)$. This means that $v_\theta^* \in \mathbb{R}^{\kappa_2}$ satisfies $v_\theta^* \in \arg\min_v \|T_\theta[\Phi v] - \Phi v\|_{d_\theta}^2$. By the projection theorem on Hilbert spaces, the orthogonality condition for $v_\theta^*$ becomes

$$\sum_{x \in \mathcal{X}} d_\theta(x|x_0)\phi(x)\phi(x)^\top v_\theta^* = \sum_{x \in \mathcal{X}} d_\theta(x|x_0)\phi(x)C_\theta(x)$$
$$+ \gamma \sum_{x \in \mathcal{X}} d_\theta(x|x_0)\phi(x) \max_{\xi\,:\,\xi P_\theta(\cdot|x) \in \mathcal{U}(x, P_\theta(\cdot|x))} \mathbb{E}_\xi[\Phi v_\theta^*].$$

As a result, given a long enough trajectory $x_0, a_0, x_1, a_1, \ldots, x_{N-1}, a_{N-1}$ generated by policy $\theta$, we may estimate the fixed-point solution $v_\theta^*$ using the projected risk sensitive value iteration (PRSVI) algorithm with the update rule

$$v_{k+1} = \left( \frac{1}{N} \sum_{t=0}^{N-1} \phi(x_t)\phi(x_t)^\top \right)^{-1} \left[ \frac{1}{N} \sum_{t=0}^{N-1} \phi(x_t)C_\theta(x_t) \right.$$
$$\left. + \gamma \frac{1}{N} \sum_{t=0}^{N-1} \phi(x_t) \max_{\xi P_\theta(\cdot|x_t) \in \mathcal{U}(x_t, P_\theta(\cdot|x_t))} \mathbb{E}_\xi[\Phi v_k] \right]. \tag{25}$$

Note that using the law of large numbers, as both $N$ and $k$ tend to infinity, $v_k$ converges w.p. 1 to $v_\theta^*$, the unique solution of the fixed point equation $\Pi T_\theta[\Phi v] = \Phi v$.

In order to implement the iterative algorithm (25), one must repeatedly solve the inner optimization problem $\max_{\xi P_\theta(\cdot|x) \in \mathcal{U}(x, P_\theta(\cdot|x))} \mathbb{E}_\xi[\Phi v]$. When the state space $\mathcal{X}$ is large, solving this optimization problem is often computationally expensive or even intractable. Similar to Section 3.4 of [37], we propose the following SAA approach to solve this problem. For the trajectory, $x_0, a_0, x_1, a_1, \ldots, x_{N-1}, a_{N-1}$, we define the empirical transition probability $P_N(x'|x,a) \doteq$

$\frac{\sum_{t=0}^{N-1} \mathbf{1}\{x_t=x,\, a_t=a,\, x_{t+1}=x'\}}{\sum_{t=0}^{N-1} \mathbf{1}\{x_t=x,\, a_t=a\}}$[4] and $P_{\theta;N}(x'|x) = \sum_{a \in \mathcal{A}} P_N(x'|x,a)\mu_\theta(a|x)$. Consider the following $\ell_2$-regularized empirical robust optimization problem[5]

$$\rho_N(\Phi v) = \max_{\xi : \xi P_{\theta;N} \in \mathcal{U}(x, P_{\theta;N})} \sum_{x' \in \mathcal{X}} P_{\theta;N}(x'|x)\xi(x')\phi^\top(x')v$$

$$+ \frac{1}{2N}\big[P_{\theta;N}(x'|x)\xi(x')\big]^2. \tag{26}$$

As in [20], the $\ell_2$-regularization term in this optimization problem guarantees convergence of optimizers $\xi^*$ and the corresponding KKT multipliers, when $N \to \infty$. Convergence of these parameters is crucial for the policy gradient analysis in the next sections. We denote by $\xi^*_{\theta,x;N}$, the solution of the above empirical optimization problem, and by $\lambda^{*,\mathcal{P}}_{\theta,x;N}$, $\lambda^{*,\mathcal{E}}_{\theta,x;N}$, $\lambda^{*,\mathcal{I}}_{\theta,x;N}$, the corresponding KKT multipliers.

We obtain the empirical PRSVI algorithm by replacing the inner optimization $\max_{\xi P_\theta(\cdot|x_t) \in \mathcal{U}(x_t, P_\theta(\cdot|x_t))} \mathbb{E}_\xi[\Phi v^*_\theta]$ in Eq. 25 with $\rho_N(\Phi v)$ from Eq. 26. Similarly, as both $N$ and $k$ tend to infinity, $v_k$ converges w.p. 1 to $v^*_\theta$. More details can be found in the supplementary material.

### F.4 Gradient Estimation

In Section F.3, we showed that we may effectively approximate the value function of a fixed policy $\theta$ using the (empirical) PRSVI algorithm in Eq. 25. In this section, we first derive a formula for the gradient of the Markov-coherent dynamic risk measure $\rho_\infty(\mathcal{M})$, and then propose a SAA algorithm for estimating this gradient, in which we use the SAA approximation of value function from Section F.3. As described in Section F.2, $\rho_\infty(\mathcal{M}) = V_\theta(x_0)$, and thus, we shall first derive a formula for $\nabla_\theta V_\theta(x_0)$.

Let $(\xi^*_{\theta,x}, \lambda^{*,\mathcal{P}}_{\theta,x}, \lambda^{*,\mathcal{E}}_{\theta,x}, \lambda^{*,\mathcal{I}}_{\theta,x})$ be the saddle point of (6) corresponding to the state $x \in \mathcal{X}$. In many common coherent risk measures such as CVaR and mean semi-deviation, there are closed-form formulas for $\xi^*_{\theta,x}$ and KKT multipliers $(\lambda^{*,\mathcal{P}}_{\theta,x}, \lambda^{*,\mathcal{E}}_{\theta,x}, \lambda^{*,\mathcal{I}}_{\theta,x})$. We will briefly discuss the case when the saddle point does not have an explicit solution later in this section. Before analyzing the gradient estimation, we have the following standard assumption in analogous to Assumption 4.1 of the static case.

**Assumption F.3.** *The likelihood ratio $\nabla_\theta \log \mu_\theta(a|x)$ is well-defined and bounded for all $x \in \mathcal{X}$ and $a \in \mathcal{A}$.*

As in Theorem 4.2 for the static case, we may use the envelope theorem and the risk-sensitive Bellman equation, $V_\theta(x) = C_\theta(x) + \gamma \max_{\xi P_\theta(\cdot|x) \in \mathcal{U}(x, P_\theta(\cdot|x))} \mathbb{E}_\xi[V_\theta]$, to derive a formula for $\nabla_\theta V_\theta(x)$. We report this result in Theorem F.4, which is analogous to the risk-neutral policy gradient theorem [34, 17, 7]. The proof is in the supplementary material.

**Theorem F.4.** *Under Assumptions 2.2, we have*

$$\nabla V_\theta(x) = \mathbb{E}_{\xi^*_\theta}\left[\sum_{t=0}^\infty \gamma^t \nabla_\theta \log \mu_\theta(a_t|x_t) h_\theta(x_t, a_t) \,\big|\, x_0 = x\right],$$

*where $\mathbb{E}_{\xi^*_\theta}[\cdot]$ denotes the expectation w.r.t. trajectories generated by a Markov chain with transition probabilities $P_\theta(\cdot|x)\xi^*_{\theta,x}(\cdot)$, and the stage-wise cost function $h_\theta(x,a)$ is defined as*

$$h_\theta(x,a) = C(x,a) + \sum_{x' \in \mathcal{X}} P(x'|x,a)\xi^*_{\theta,x}(x')\Big[\gamma V_\theta(x') - \lambda^{*,\mathcal{P}}_{\theta,x}$$

$$- \sum_{i \in \mathcal{I}} \lambda^{*,\mathcal{I}}_{\theta,x}(i) \frac{df_i(\xi^*_{\theta,x}, p)}{dp(x')} - \sum_{e \in \mathcal{E}} \lambda^{*,\mathcal{E}}_{\theta,x}(e) \frac{dg_e(\xi^*_{\theta,x}, p)}{dp(x')}\Big]. \tag{27}$$

Theorem F.4 indicates that the policy gradient of the Markov-coherent dynamic risk measure $\rho_\infty(\mathcal{M})$, i.e., $\nabla_\theta \rho_\infty(\mathcal{M}) = \nabla_\theta V_\theta$, is equivalent to the risk-neutral value function of policy $\theta$ in a MDP with the stage-wise cost function $\nabla_\theta \log \mu_\theta(a|x)h_\theta(x,a)$ (which is well-defined and bounded), and transition probability $P_\theta(\cdot|x)\xi^*_{\theta,x}(\cdot)$. Thus, when the saddle points are known and the state space $\mathcal{X}$ is not too large, we can compute $\nabla_\theta V_\theta$ using a policy evaluation algorithm. However, when the state space is large, exact calculation of $\nabla V_\theta$ by policy evaluation becomes impossible, and our goal would be to derive a sampling method to estimate $\nabla V_\theta$. Unfortunately, since the risk envelop depends on the policy parameter $\theta$, unlike the risk-neutral case, the risk sensitive (or robust) Bellman equation $T_\theta[V_\theta](x)$ in (23) is nonlinear in the stationary Markov policy $\mu_\theta$. Therefore $h_\theta$ cannot be considered using the action-value function ($Q$-function) of the robust MDP. Therefore, even if the exact formulation of the value function $V_\theta$ is known, it is computationally intractable to enumerate the summation over $x'$ to compute $h_\theta(x,a)$. On top of that in many applications the value function $V_\theta$ is not known in advance, which further complicates gradient estimation. To estimate the policy gradient when the value function is unknown, we approximate it by the projected risk sensitive value function $\Phi v^*_\theta$. To address the sampling issues, we propose the following *two-phase sampling procedure* for estimating $\nabla V_\theta$.

**(1)** Generate $N$ trajectories $\{x_0^{(j)}, a_0^{(j)}, x_1^{(j)}, a_1^{(j)}, \ldots\}_{j=1}^N$ from the Markov chain induced by policy $\theta$ and transition probabilities $P_\theta^\xi(\cdot|x) := \xi^*_{\theta,x}(\cdot)P_\theta(\cdot|x)$.

**(2)** For each state-action pair $(x_t^{(j)}, a_t^{(j)}) = (x,a)$, generate $N$ samples $\{y^{(k)}\}_{k=1}^N$ using the transition probability $P(\cdot|x,a)$ and calculate the following empirical average estimate of $h_\theta(x,a)$

$$
h_{\theta,N}(x,a) := C(x,a) + \frac{1}{N}\sum_{k=1}^N \xi^*_{\theta,x}(y^{(k)})\Bigg[\gamma v^{*\top}_\theta \phi(y^{(k)}) - \lambda^{*,\mathcal{P}}_{\theta,x}
$$
$$
- \sum_{i\in\mathcal{I}} \lambda^{*,\mathcal{I}}_{\theta,x}(i)\frac{df_i(\xi^*_{\theta,x},p)}{dp(y^{(k)})} - \sum_{e\in\mathcal{E}} \lambda^{*,\mathcal{E}}_{\theta,x}(e)\frac{dg_e(\xi^*_{\theta,x},p)}{dp(y^{(k)})}\Bigg]
$$

**(3)** Calculate an estimate of $\nabla V_\theta$ using the following average over all the samples: $\frac{1}{N}\sum_{j=1}^N \sum_{t=0}^\infty \gamma^t \nabla_\theta \log \mu_\theta(a_t^{(j)}|x_t^{(j)})h_{\theta,N}(x_t^{(j)}, a_t^{(j)})$.

Indeed, by the definition of empirical transition probability $P_N(x'|x,a)$, $h_{\theta,N}(x,a)$ can be re-written as in the same structure of $h_\theta(x,a)$, except by replacing the transition probability $P(x'|x,a)$ with $P_N(x'|x,a)$.

Furthermore, in the case that the saddle points $(\xi^*_{\theta,x}, \lambda^{*,\mathcal{P}}_{\theta,x}, \lambda^{*,\mathcal{E}}_{\theta,x}, \lambda^{*,\mathcal{I}}_{\theta,x})$ do not have a closed-form solution, we may follow the SAA procedure of Section F.3 and replace them and the transition probabilities $P(x'|x,a)$ with their sample estimates $(\xi^*_{\theta,x;N}, \lambda^{*,\mathcal{P}}_{\theta,x;N}, \lambda^{*,\mathcal{E}}_{\theta,x;N}, \lambda^{*,\mathcal{I}}_{\theta,x;N})$ and $P_N(x'|x,a)$ respectively.

At the end, we show the convergence of the above two-phase sampling procedure. Let $d_{P_\theta^\xi}(x|x_0)$ and $\pi_{P_\theta^\xi}(x,a|x_0)$ be the state and state-action occupancy measure induced by the transition probability function $P_\theta^\xi(\cdot|x)$, respectively. Similarly, let $d_{P_{\theta;N}^\xi}(x|x_0)$ and $\pi_{P_{\theta;N}^\xi}(x,a|x_0)$ be the state and state-action occupancy measure induced by the estimated transition probability function $P_{\theta;N}^\xi(\cdot|x) := \xi^*_{\theta,x;N}(\cdot)P_{\theta;N}(\cdot|x)$. From the two-phase sampling procedure for policy gradient estimation and by the strong law of large numbers, when $N \to \infty$, with probability 1, we have that $\frac{1}{N}\sum_{j=1}^N \sum_{t=0}^\infty \gamma^t \mathbf{1}\{x_t^{(j)} = x, a_t^{(j)} = a\} = \pi_{P_{\theta;N}^\xi}(x,a|x_0)$. Based on the strongly convex property of the $\ell_2$-regularized objective function in the inner robust optimization problem $\rho_N(\Phi v)$, we can show that both the state-action occupancy measure $\pi_{P_{\theta;N}^\xi}(x,a|x_0)$ and the stage-wise cost $h_{\theta;N}(x,a)$ converge to the their true values within a value function approximation error bound $\Delta = \|\Phi v^*_\theta - V_\theta\|_\infty$. We refer the readers to the supplementary materials for these technical results. These results together with Theorem F.4 imply the consistency of the policy gradient estimation.

**Theorem F.5.** *For any $x_0 \in \mathcal{X}$, the following expression holds with probability 1:*

$$\left| \lim_{N \to \infty} \frac{1}{N} \sum_{j=1}^{N} \sum_{t=0}^{\infty} \gamma^t \, \nabla \log \mu_\theta(a_t^{(j)} | x_t^{(j)}) \, h_{\theta,N}(x_t^{(j)}, a_t^{(j)}) \right.$$

$$\left. - \nabla V_\theta(x_0) \right| = O(\Delta).$$

Thm. F.5 guarantees that as the value function approximation error decreases and the number of samples increases, the sampled gradient converges to the true gradient.

# G  Convergence Analysis of Empirical PRSVI

**Lemma G.1** (Technical Lemma)**.** *Let $P(\cdot|\cdot)$ and $\widetilde{P}(\cdot|\cdot)$ be two arbitrary transition probability matrices. At state $x \in \mathcal{X}$, for any $\xi : \xi P(\cdot|x) \in \mathcal{U}(x, P(\cdot|x))$, there exists a $M_\xi > 0$ such that for some $\tilde{\xi} : \tilde{\xi} \widetilde{P}(\cdot|x) \in \mathcal{U}(x, \widetilde{P}(\cdot|x))$,*

$$\sum_{x' \in \mathcal{X}} |\xi(x') - \tilde{\xi}(x')| \le M_\xi \sum_{x' \in \mathcal{X}} \left| P(x'|x) - \widetilde{P}(x'|x) \right|.$$

*Proof.* From Theorem 2.1, we know that $\mathcal{U}(x, P(\cdot|x))$ is a closed, bounded, convex set of probability distribution functions. Since any conditional probability mass function $P$ is in the interior of $\text{dom}(\mathcal{U})$ and the graph of $\mathcal{U}(x, P(\cdot|x))$ is closed, by Theorem 2.7 in [29], $\mathcal{U}(x, P(\cdot|x))$ is a Lipschitz set-valued mapping with respect to the Hausdorff distance. Thus, for any $\xi : \xi P(\cdot|x) \in \mathcal{U}(x, P(\cdot|x))$, the following expression holds for some $M_\xi > 0$:

$$\inf_{\hat{\xi} \in \mathcal{U}(x, \widetilde{P}(\cdot|x))} \sum_{x' \in \mathcal{X}} |\xi(x') - \hat{\xi}(x')| \le M_\xi \sum_{x' \in \mathcal{X}} \left| P(x'|x) - \widetilde{P}(x'|x) \right|.$$

Next, we want to show that the infimum of the left side is attained. Since the objective function is convex, and $\mathcal{U}(x, \widetilde{P}(\cdot|x))$ is a convex compact set, there exists $\tilde{\xi} : \tilde{\xi} \widetilde{P}(\cdot|x) \in \mathcal{U}(x, \widetilde{P}(\cdot|x))$ such that infimum is attained. $\square$

**Lemma G.2** (Strong Law of Large Number)**.** *Consider the sampling based PRSVI algorithm with update sequence $\{\widehat{v}_k\}$. Then as both $N$ and $k$ tend to $\infty$, $\widehat{v}_k$ converges with probability 1 to $v_\theta^*$, the unique solution of projected risk sensitive fixed point equation $\Pi T_\mu[\Phi v] = \Phi v$.*

*Proof.* By the strong law of large number of Markov process, the empirical visiting distribution and transition probability asymptotically converges to their statistical limits with probability 1, i.e.,

$$\frac{\sum_{t=0}^{N-1} \mathbf{1}\{x_t = x\}}{N} \to d_\theta(x|x_0), \text{ and } \widehat{P}(x'|x,a) \to P(x'|x,a), \forall x, x' \in \mathcal{X}, a \in \mathcal{A}.$$

Therefore with probability 1,

$$\frac{1}{N} \sum_{t=0}^{N-1} \phi(x_t)\phi(x_t)^\top \to \sum_x d_\theta(x|x_0) \cdot \phi(x)\phi^\top(x),$$

$$\frac{1}{N} \sum_{t=0}^{N-1} \phi(x_t)C_\theta(x_t) \to \sum_x d_\theta(x|x_0) \cdot \phi(x)C_\theta(x).$$

Now we show that following expression holds with probability 1:

$$\max_{\xi : \xi P_{\theta;N}(\cdot|x_t) \in \mathcal{U}(x_t, P_{\theta;N}(\cdot|x_t))} \sum_{x' \in \mathcal{X}} \xi(x') P_{\theta;N}(x'|x_t) v^\top \phi(x') + \frac{1}{2N} (\xi(x') P_{\theta;N}(x'|x_t))^2$$

$$\to \max_{\xi : \xi P_\theta(\cdot|x_t) \in \mathcal{U}(x_t, P_\theta(\cdot|x_t))} \sum_{x' \in \mathcal{X}} \xi(x') P_\theta(x'|x_t) v^\top \phi(x'). \tag{28}$$

Notice that for $\{\xi_{\theta,x_t;N}^*(x')\}_{x'\in\mathcal{X}} \in \arg\max_{\xi\,:\,\xi P_{\theta;N}(\cdot|x_t)\in\mathcal{U}(x_t,P_{\theta;N}(\cdot|x_t))} \sum_{x'\in\mathcal{X}} \xi(x')P_{\theta;N}(x'|x_t)v^\top\phi(x')$,
Lemma G.1 implies

$$\max_{\xi\,:\,\xi P_{\theta;N}(\cdot|x_t)\in\mathcal{U}(x_t,P_{\theta;N}(\cdot|x_t))} \sum_{x'\in\mathcal{X}} \xi(x')P_{\theta;N}(x'|x_t)v^\top\phi(x') + \frac{1}{2N}(\xi(x')P_{\theta;N}(x'|x_t))^2$$

$$- \max_{\xi\,:\,\xi P_\theta(\cdot|x_t)\in\mathcal{U}(x_t,P_\theta(\cdot|x_t))} \sum_{x'\in\mathcal{X}} \xi(x')P_\theta(x'|x_t)v^\top\phi(x')$$

$$\leq \|\Phi v\|_\infty \left( M_{\xi_{\theta,x_t;N}^*} + \max_{x\in\mathcal{X}} |\xi_{\theta,x_t;N}^*(x)| \right) \sum_{x'\in\mathcal{X}} |P_\theta(x'|x_t) - P_{\theta;N}(x'|x_t)| + \frac{1}{2N}.$$

The quantity $\max_{x\in\mathcal{X}} |\xi_{\theta,x_t;N}^*(x)|$ is bounded because $\mathcal{U}(x_t, P_{\theta;N}(\cdot|x_t))$ is a closed and bounded convex set from the definition of coherent risk measures. By repeating the above analysis by interchanging $P_\theta$ and $P_{\theta;N}$ and combining previous arguments, one obtains

$$\left| \max_{\xi\,:\,\xi P_{\theta;N}(\cdot|x_t)\in\mathcal{U}(x_t,P_{\theta;N}(\cdot|x_t))} \sum_{x'\in\mathcal{X}} \xi(x')P_{\theta;N}(x'|x_t)v^\top\phi(x') + \frac{1}{2N}(\xi(x')P_{\theta;N}(x'|x_t))^2 \right.$$

$$\left. - \max_{\xi\,:\,\xi P_\theta(\cdot|x_t)\in\mathcal{U}(x_t,P_\theta(\cdot|x_t))} \sum_{x'\in\mathcal{X}} \xi(x')P_\theta(x'|x_t)v^\top\phi(x') \right|$$

$$\leq \|\Phi v\|_\infty \max\left\{ \left( M_{\xi^*} + \max_{x\in\mathcal{X}} |\xi^*(x)| \right), \left( M_{\xi_{\theta,x_t;N}^*} + \max_{x\in\mathcal{X}} |\xi_{\theta,x_t;N}^*(x)| \right) \right\} \sum_{x'\in\mathcal{X}} |P_\theta(x'|x_t) - P_{\theta;N}(x'|x_t)| + \frac{1}{2N}.$$

Therefore, the claim in expression (28) holds when $N \rightarrow \infty$ and $\sum_{x'\in\mathcal{X}} |P_\theta(x'|x_t) - P_{\theta;N}(x'|x_t)| \rightarrow 0$. On the other hand, the strong law of large numbers also implies that with probability 1,

$$\frac{1}{N}\sum_{t=0}^{N-1} \phi(x_t)\rho(\Phi v_t) \rightarrow d_\theta(x|x_0)\phi(x) \max_{\xi\,:\,\xi P_\theta(\cdot|x)\in\mathcal{U}(x,P_\theta(\cdot|x))} \sum_{x'\in\mathcal{X}} \xi(x')P_\theta(x'|x)v_\theta^{*\top}\phi(x').$$

Combining the above arguments implies

$$\frac{1}{N}\sum_{t=0}^{N-1} \phi(x_t)\rho_N(\Phi v_t) \rightarrow d_\theta(x|x_0)\phi(x) \max_{\xi\,:\,\xi P_\theta(\cdot|x)\in\mathcal{U}(x,P_\theta(\cdot|x))} \sum_{x'\in\mathcal{X}} \xi(x')P_\theta(x'|x)v_\theta^{*\top}\phi(x').$$

As $N \rightarrow \infty$, the above arguments imply that $v_k - \widehat{v}_k \rightarrow 0$. On the other hand, Proposition 1 in [37] implies that the projected risk sensitive Bellman operator $\Pi T_\theta[V]$ is a contraction, it follows that from the analysis in Section 6.3 in [5] that the sequence $\{\Phi\widehat{v}_k\}$ generated by projected value iteration converges to the unique fixed point $\Phi v_\theta^*$. This in turns implies that the sequence $\{\Phi v_k\}$ converges to $\Phi v_\theta^*$. $\qquad\square$

## H  Technical Results

Since by convention $\xi_{\theta,x;N}^*(x') = 0$ whenever $P_{\theta;N}(x'|x) = 0$. In this section, we simplify the analysis by letting $P_{\theta;N}(x'|x) > 0$ for any $x' \in \mathcal{X}$ without loss of generality. Consider the following empirical robust optimization problem:

$$\max_{\xi\,:\,\xi P_{\theta;N}(\cdot|x)\in\mathcal{U}(x,P_{\theta;N}(\cdot|x))} \sum_{x'\in\mathcal{X}} P_{\theta;N}(x'|x)\xi(x')V_\theta(x'), \tag{29}$$

where the solution of the above empirical problem is $\bar{\xi}_{\theta,x;N}^*$ and the corresponding KKT multipliers are $(\bar{\lambda}_{\theta,x;N}^{*,\mathcal{P}}, \bar{\lambda}_{\theta,x;N}^{*,\mathcal{E}}, \bar{\lambda}_{\theta,x;N}^{*,\mathcal{I}})$. Comparing to the optimization problem for $\rho_N(\Phi v)$, i.e.,

$$\rho_N(\Phi v) = \max_{\xi\,:\,\xi P_{\theta;N}(\cdot|x)\in\mathcal{U}(x,P_{\theta;N}(\cdot|x))} \sum_{x'\in\mathcal{X}} P_{\theta;N}(x'|x)\xi(x')\phi^\top(x')v + \frac{1}{2N}(\xi(x')P_{\theta;N}(x'|x))^2,$$

$$\tag{30}$$

where the solution of the above empirical problem is $\xi^*_{\theta,x;N}$ and the corresponding KKT multipliers are $(\lambda^{*,\mathcal{P}}_{\theta,x;N}, \lambda^{*,\mathcal{E}}_{\theta,x;N}, \lambda^{*,\mathcal{I}}_{\theta,x;N})$, the optimization problem in (29) can be viewed as having a skewed objective function of the problem in (30), within the deviation of magnitude $\Delta + 1/2N$ where $\Delta = \|\Phi v^*_\theta - V_\theta\|_\infty$. Before getting into the main analysis, we have the following observations.

**(i)** Without loss of generality, we can also assume $(\xi^*_{\theta,x;N}, (\lambda^{*,\mathcal{P}}_{\theta,x;N}, \lambda^{*,\mathcal{E}}_{\theta,x;N}, \lambda^{*,\mathcal{I}}_{\theta,x;N}))$ follows the strict complementary slackness condition[6].

**(ii)** Recall from Assumption 2.2 that the functions $f_i(\xi, p)$ and $g_e(\xi, p)$ are twice differentiable in $\xi$ at $p = P_{\theta,N}(\cdot|x)$ for any $x \in \mathcal{X}$.

**(iii)** The Slater's condition in Assumption 2.2 implies the linear independence constraint qualification (LICQ).

**(iv)** Since optimization problem (30) has a convex objective function and convex/affine constraints in $\xi \in \mathbb{R}^{|\mathcal{X}|}$, equipped with the Slater's condition we have that the first order KKT condition holds at $\xi^*_{\theta,x;N}$ with the corresponding KKT multipliers are $(\lambda^{*,\mathcal{P}}_{\theta,x;N}, \lambda^{*,\mathcal{E}}_{\theta,x;N}, \lambda^{*,\mathcal{I}}_{\theta,x;N})$. Furthermore, define the Lagrangian function

$$\widehat{L}_{\theta;N}(\xi, \lambda^{\mathcal{P}}, \lambda^{\mathcal{E}}, \lambda^{\mathcal{I}}) \doteq \sum_{x'\in\mathcal{X}} P_{\theta;N}(x'|x)\xi(x')\phi^\top(x')v + \frac{1}{2N}(P_{\theta;N}(x'|x)\xi(x'))^2$$

$$- \lambda^{\mathcal{P}}\left(\sum_{x'\in\mathcal{X}}\xi(x')P_{\theta;N}(x'|x) - 1\right)$$

$$- \sum_{e\in\mathcal{E}}\lambda^{\mathcal{E}}(e)f_e(\xi, P_{\theta;N}(\cdot|x)) - \sum_{i\in\mathcal{I}}\lambda^{\mathcal{I}}(i)f_i(\xi, P_{\theta;N}(\cdot|x)).$$

One can easily conclude that $\nabla^2\widehat{L}_{\theta;N}(\xi, \lambda^{\mathcal{P}}, \lambda^{\mathcal{E}}, \lambda^{\mathcal{I}}) = -P_{\theta;N}(\cdot|x)^\top P_{\theta;N}(\cdot|x)/N - \sum_{i\in\mathcal{I}}\lambda^{\mathcal{I}}(i)\nabla^2_\xi f_i(\xi, P_{\theta;N}(\cdot|x))$ such that for any vector $\nu \neq 0$,

$$\nu^\top\nabla^2\widehat{L}_{\theta;N}(\xi^*_{\theta,x;N}, \lambda^{*,\mathcal{P}}_{\theta,x;N}, \lambda^{*,\mathcal{E}}_{\theta,x;N}, \lambda^{*,\mathcal{I}}_{\theta,x;N})\nu < 0,$$

which further implies that the second order sufficient condition (SOSC) holds at $(\xi^*_{\theta,x;N}, \lambda^{*,\mathcal{P}}_{\theta,x;N}, \lambda^{*,\mathcal{E}}_{\theta,x;N}, \lambda^{*,\mathcal{I}}_{\theta,x;N})$.

Based on all the above analysis, we have the following sensitivity result from Corollary 3.2.4 in [13], derived based on Implicit Function Theorem.

**Proposition H.1** (Basic Sensitivity Theorem). *Under the Assumption 2.2, for any $x \in \mathcal{X}$ there exists a bounded non-singular matrix $K_{\theta,x}$ and a bounded vector $L_{\theta,x}$, such that the difference between the optimizers and KKT multipliers of optimization problem (29) and (30) are bounded as follows:*

$$\begin{bmatrix}\bar{\xi}^*_{\theta,x;N}\\ \bar{\lambda}^{*,\mathcal{I}}_{\theta,x;N}\\ \bar{\lambda}^{*,\mathcal{P}}_{\theta,x;N}\\ \bar{\lambda}^{*,\mathcal{E}}_{\theta,x;N}\end{bmatrix} = \begin{bmatrix}\xi^*_{\theta,x;N}\\ \lambda^{*,\mathcal{I}}_{\theta,x;N}\\ \lambda^{*,\mathcal{P}}_{\theta,x;N}\\ \lambda^{*,\mathcal{E}}_{\theta,x;N}\end{bmatrix} + \Phi^{-1}_{\theta,x}\Psi_{\theta,x}\left(\Delta + \frac{1}{2N}\right) + o\left(\Delta + \frac{1}{2N}\right).$$

On the other hand, we know from Proposition 4.4 that $\bar{\xi}^*_{\theta,x;N} \to \xi^*_{\theta,x}$ and $(\bar{\lambda}^{*,\mathcal{P}}_{\theta,x;N}, \bar{\lambda}^{*,\mathcal{E}}_{\theta,x;N}, \bar{\lambda}^{*,\mathcal{I}}_{\theta,x;N}) \to (\lambda^{*,\mathcal{P}}_{\theta,x}, \lambda^{*,\mathcal{E}}_{\theta,x}, \lambda^{*,\mathcal{I}}_{\theta,x})$ with probability 1 as $N \to \infty$. Also recall from the law of large numbers that the sampled approximation error $\max_{x\in\mathcal{X},a\in\mathcal{A}}\|P(\cdot|x,a) - P_N(\cdot|x,a)\|_1 \to 0$ almost surely as $N \to \infty$. Then we have the following error bound in the stage-wise cost approximation $\widehat{h}_{\theta;N}(x,a)$ and $\gamma-$visiting distribution $\pi_N(x,a)$.

**Lemma H.2.** *There exists a constant $M_h > 0$ such that $\max_{x\in\mathcal{X},a\in\mathcal{A}}|h_\theta(x,a) - \lim_{N\to\infty}\widehat{h}_{\theta;N}(x,a)| \leq M_h\Delta$.*

*Proof.* First we can easily see that for any state $x \in \mathcal{X}$ and action $a \in \mathcal{A}$,

$$|\widehat{h}_{\theta;N}(x,a) - h_\theta(x,a)| \leq M \sum_{i \in \mathcal{I}} \left| \lambda_{\theta,x;N}^{*,\mathcal{I}}(i) - \lambda_{\theta,x}^{*,\mathcal{I}}(i) \right| + M \sum_{e \in \mathcal{E}} \left| \lambda_{\theta,x;N}^{*,\mathcal{E}}(e) - \lambda_{\theta,x}^{*,\mathcal{E}}(e) \right| + \left| \lambda_{\theta,x;N}^{*,\mathcal{P}} - \lambda_{\theta,x}^{*,\mathcal{P}} \right|$$
$$+ \gamma \|V_\theta\|_\infty \|\xi_{\theta,x;N}^* - \xi_{\theta,x}^*\|_1 + \gamma \|V_\theta - \Phi v_\theta^*\|_\infty$$
$$+ \gamma \|V_\theta\|_\infty \max\{\|\xi_{\theta,x;N}^*\|_\infty, \|\xi_{\theta,x}^*\|_\infty\} \|P(\cdot|x,a) - P_N(\cdot|x,a)\|_1.$$

Note that at $N \to \infty$, $\|P(\cdot|x,a) - P_N(\cdot|x,a)\|_1 \to 0$ with probability 1. Both $\|\xi_{\theta;N}^*\|_\infty$ and $\|\xi_{\theta,x}^*\|_\infty$ are finite valued because $\mathcal{U}(P_\theta)$ and $\mathcal{U}(P_{\theta;N})$ are convex compact sets of real vectors. Therefore, by noting that $\|V_\theta\|_\infty \leq C_{\max}/(1-\gamma)$ and applying Proposition 4.4 and H.1, the proof of this Lemma is completed by letting $N \to \infty$ and defining

$$M_h(x) = \max\{1, M, \frac{\gamma C_{\max}}{1-\gamma}\} \left\| \begin{bmatrix} \xi_{\theta,x;N}^* - \bar{\xi}_{\theta,x;N}^* \\ \lambda_{\theta,x;N}^{*,\mathcal{I}} - \bar{\lambda}_{\theta,x;N}^{*,\mathcal{I}} \\ \lambda_{\theta,x;N}^{*,\mathcal{P}} - \bar{\lambda}_{\theta,x;N}^{*,\mathcal{P}} \\ \lambda_{\theta,x;N}^{*,\mathcal{E}} - \bar{\lambda}_{\theta,x;N}^{*,\mathcal{E}} \end{bmatrix} + \begin{bmatrix} \bar{\xi}_{\theta,x;N}^* - \xi_{\theta,x}^* \\ \bar{\lambda}_{\theta,x;N}^{*,\mathcal{I}} - \lambda_{\theta,x}^{*,\mathcal{I}} \\ \bar{\lambda}_{\theta,x;N}^{*,\mathcal{P}} - \lambda_{\theta,x}^{*,\mathcal{P}} \\ \bar{\lambda}_{\theta,x;N}^{*,\mathcal{E}} - \lambda_{\theta,x}^{*,\mathcal{E}} \end{bmatrix} \right\|_1 + \gamma \Delta$$

$$\leq \left( \max\{1, M, \frac{\gamma C_{\max}}{1-\gamma}\} \|\Phi_{\theta,x}^{-1} \Psi_{\theta,x}\|_1 + \gamma \right) \Delta.$$

$\square$

**Lemma H.3.** *There exists a constant $M_\pi > 0$ such that $\|\pi - \lim_{N\to\infty} \pi_N\|_1 \leq M_\pi \Delta$.*

*Proof.* First, recall that the $\gamma-$visiting distribution satisfies the following identity:

$$\gamma \sum_{x' \in \mathcal{X}} d_{P_\theta^\xi}(x'|x) P_\theta^\xi(x|x') = d_{P_\theta^\xi}(x) - (1-\gamma)\mathbf{1}\{x_0 = x\}, \tag{31}$$

From here one easily notice this expression can be rewritten as follows:

$$\left( I - \gamma P_\theta^\xi \right)^\top d_{P_\theta^\xi}(\cdot|x) = \mathbf{1}\{x_0 = x\}, \ \forall x \in \mathcal{X}.$$

On the other hand, by repeating the analysis with $P_{\theta;N}(\cdot|x)$, we can also write

$$\left( I - \gamma P_{\theta;N}^\xi \right)^\top d_{P_{\theta;N}^\xi} = \{\mathbf{1}\{x_0 = z\}\}_{z \in \mathcal{X}}.$$

Combining the above expressions implies for any $x \in \mathcal{X}$,

$$d_{P_\theta^\xi} - d_{P_{\theta;N}^\xi} - \gamma \left( \left( P_\theta^\xi \right)^\top d_{P_\theta^\xi} - (P_{\theta;N}^\xi)^\top d_{P_{\theta;N}^\xi} \right) = 0,$$

which further implies

$$\left( I - \gamma P_\theta^\xi \right)^\top \left( d_{P_\theta^\xi} - d_{P_{\theta;N}^\xi} \right) = \gamma \left( P_\theta^\xi - P_{\theta;N}^\xi \right)^\top d_{P_{\theta;N}^\xi}$$

$$\iff \left( d_{P_\theta^\xi} - d_{P_{\theta;N}^\xi} \right) = \left( I - \gamma P_\theta^\xi \right)^{-\top} \gamma \left( P_\theta^\xi - P_{\theta;N}^\xi \right)^\top d_{P_{\theta;N}^\xi}.$$

Notice that with transition probability matrix $P_\theta^\xi(\cdot|x)$, we have $(I - \gamma P_\theta^\xi)^{-1} = \sum_{t=0}^\infty \left( \gamma P_\theta^\xi \right)^k < \infty$. The series is summable because by Perron-Frobenius theorem, the maximum eigenvalue of $P_\theta^\xi$ is less than or equal to 1 and $I - \gamma P_\theta^\xi$ is invertible. On the other hand, for every given $x_0 \in \mathcal{X}$,

$$\left\{ \left( P_\theta^\xi - P_{\theta;N}^\xi \right)^\top d_{P_{\theta;N}^\xi} \right\}(z') = \sum_{x \in \mathcal{X}} \sum_{k=0}^\infty \gamma^k (1-\gamma) \mathbb{P}_{P_{\theta;N}^\xi}(x_k = x|x_0) \left( P_\theta^\xi(z'|x) - P_{\theta;N}^\xi(z'|x) \right), \ \forall z' \in \mathcal{X}$$

$$= \mathbb{E}_{P_{\theta;N}^\xi} \left( \sum_{k=0}^\infty \gamma^k (1-\gamma) \left( P_\theta^\xi(z'|x_k) - P_{\theta;N}^\xi(z'|x_k) \right) | x_0 \right), \ \forall z' \in \mathcal{X}$$

$$\leq \mathbb{E}_{P_{\theta;N}^\xi} \left( \sum_{k=0}^\infty \gamma^k (1-\gamma) \left| P_\theta^\xi(z'|x_k) - P_{\theta;N}^\xi(z'|x_k) \right| | x_0 \right), \ \forall z' \in \mathcal{X}$$

$$\doteq \mathcal{Q}(z'), \ \forall z' \in \mathcal{X}.$$

Note that every element in matrix $(I - \gamma P_\theta^\xi)^{-1} = \sum_{t=0}^\infty \left(\gamma P_\theta^\xi\right)^k$ is non-negative. This implies for any $z \in \mathcal{X}$,

$$\left|\left\{d_{P_\theta^\xi} - d_{P_{\theta;N}^\xi}\right\}(z)\right| = \left|\left\{\left(I - \gamma P_\theta^\xi\right)^{-\top} \gamma \left(P_\theta^\xi - P_{\theta;N}^\xi\right)^\top d_{P_{\theta;N}^\xi}\right\}(z)\right|,$$

$$\leq \left|\left\{\left(I - \gamma P_\theta^\xi\right)^{-\top} \gamma \mathcal{Q}\right\}(z)\right| = \left\{\left(I - \gamma P_\theta^\xi\right)^{-\top} \gamma \mathcal{Q}\right\}(z).$$

The last equality is due to the fact that every element in vector $\mathcal{Q}$ is non-negative. Combining the above results with Proposition 4.4 and H.1, and noting that

$$(I - \gamma P_\theta^\xi)^{-1} e = \sum_{t=0}^\infty \left(\gamma P_\theta^\xi\right)^k e = \frac{1}{1-\gamma} e,$$

we further have that

$$\|\pi - \pi_N\|_1 = \|d_{P_\theta^\xi} - d_{P_{\theta;N}^\xi}\|_1$$

$$\leq e^\top \left(I - \gamma P_\theta^\xi\right)^{-\top} \gamma \mathcal{Q}$$

$$= \frac{\gamma}{1-\gamma} e^\top \mathcal{Q}$$

$$\leq \frac{\gamma}{1-\gamma} \max_{x \in \mathcal{X}} \left\|P_\theta^\xi(\cdot|x) - P_{\theta;N}^\xi(\cdot|x)\right\|_1$$

$$\leq \frac{\gamma}{1-\gamma} \max_{x \in \mathcal{X}} \left(\|\xi_{\theta,x}^*(\cdot) - \xi_{\theta,x;N}^*(\cdot)\|_1 \|P_\theta(\cdot|x)\|_\infty + \max\{\|\xi_{\theta,x;N}^*\|_\infty, \|\xi_{\theta,x}^*\|_\infty\} \|P(\cdot|x,a) - P_N(\cdot|x,a)\|_1\right),$$

As in previous arguments, when $N \to \infty$, one obtains $\|P(\cdot|x,a) - P_N(\cdot|x,a)\|_1 \to 0$ with probability 1 and $\|\xi_{\theta,x}^*(\cdot) - \xi_{\theta,x;N}^*(\cdot)\|_1 \to 0$. We thus set the constant $M_\pi$ as $\gamma \|\Phi_{\theta,x}^{-1} \Psi_{\theta,x}\|_1/(1-\gamma)$. $\quad\square$