[Reviews · NeurIPS 2015]

Submitted by Assigned_Reviewer_1

The paper proposes a new formula for the class of static coherent risks from which policy gradients for many common risk measures can be derived analytically. It also presents a general sampling-based algorithm for cases where no analytical solution is available and shows consistency guarantees of the policy gradient estimates. Eventually, the paper provides expressions for policy gradients of Markov coherent dynamic risk measures.

Significance and quality:

The unified derivation of policy gradients for coherent risk measures is intriguing and a valuable and novel contribution to the risk-sensitive decision making literature. Nevertheless, analytical solutions of the Lagrangian saddle-points are only available for some risk measures.

The proposed sample-based approximation scheme could allow us to apply the derived results for general coherent risk measures by avoiding to derive analytical forms of the saddle points. However, the paper does not provide any empirical evidence that the proposed estimation scheme is viable in practice. An experimental comparison between the using the sample-based policy gradients and the analytical ones in different problems, e.g. for CVaR would be extremely helpful.

Originality: To the best of my knowledge, the derived policy gradients are - in this general form - are novel.

Clarity: The paper is generally well written, however, it appears to me that the clarity of the main paper is hindered considerably by number of different issues attempted to be addressed in the eight pages. Especially the dynamic risk policy gradients in Section 5 are presented on a very high level and the lack of any algorithm details and experimental evaluations do not allow the reader to judge whether the derived policy gradients yield approaches that are viable in practice. It seems to me this content is rather suited for a longer journal publication. An alternative that I would see as beneficial is to split this paper and rather focus on static risk measures which allows to make the paper more standalone and provide more details and a more comprehensive empirical evaluation beyond the very basic toy example in the current submission.
Summary: An interesting paper with potentially generally applicable results, but the lack of comprehensive experimental evaluations makes this hard to judge. Also, a narrower focus might be more appropriate to improve the deficits in clarity due to space constraints.

Submitted by Assigned_Reviewer_2

This paper presents methods for estimating policy gradients in risk-sensitive problems with coherent risk measures, and proposes RL algorithms based on them. The paper particularly extends the existing methods for specific risk measures to the general class of coherent risk measures, and thus provides a unified view. The algorithm is illustrated with a toy example in finance domain, where each asset has its own return distribution.

The motivation is very clear, and the problem formulation and algorithms follow naturally from this motivation and previous works. The experiment is relatively simple as it is done in only a static risk problem. However, the results seem to support the quality of the algorithm in terms of its risk-aversive behavior as well as the flexibility in different risk types.

Quality: the paper is well written. The equations are technically sound and all relevant references are given.

Clarity: the clarity of this paper is well above average.

Originality: The paper proposes new algorithms for risk-sensitive RL problems based on the related works on risk-sensitive planning and RL. The extension to proposed approach is not straightforward. The originality of this paper seems sufficient to me.

Significance: Risk-sensitive RL is an interesting and important problem, and this paper provides significant contributions.
Summary: This paper presents clear algorithms which solve risk-sensitive RL with coherent risk measures, and demonstrates the behavior of trained policies with different risk settings on a static risk problem. The proposed approach is clearly motivated by the authors, and a natural extension to existing algorithms and problems (policy gradient, actor-critic and coherent risk measure optimization).

Submitted by Assigned_Reviewer_3

The authors analyze the risk averse Markov decision process setting with static and dynamically consistent risk measures (mappings). The main contribution is, in particular, to show to form of the gradient for both the dynamic and static setting.

The paper is a generalization of previous results, which focused specifically on CVaR. This extension represents a minor, but useful, contribution. The results is a simple extension of the policy gradient in risk-neutral settings.

The paper is well organized, well written, and easy to follow.

The results are correct as far as I can tell.

Minor comments:

- Theorem 2.1: The notation \xi P_\theta is confusing. I suggest referring to Theorem 6.6 in [26] and using the same notation.

- Line

141: risk enevlop[e] - Section 3: To avoid confusing readers not familiar with the topic, it may be worth pointing out here that

MDPs with Markov risk measures are very tractable. The reason the problem in the paper is not tractable is because of the policy parametrization and NOT because of the risk measure.
Summary: The paper presents a minor but solid and important extension of existing results.

Submitted by Assigned_Reviewer_4

The authors consider the problem of finding policy parameters that minimize a given risk measure. The class of risk measures considered in the paper is general, and defined by simple coherence properties such as convexity, monotonicity, and translation invariance. The presented approach is based on a dual representation of any coherent risk measures as a convex hull of probability distributions. The risk of a random variable (such as a cost) is given by the maximum expected value in the hull. The paper makes an assumption on the family of coherent risk measures, by bounding the maximum derivatives of the convex hull's borders (i.e., the borders of the hull are M-Lipschitz continuous). Using this assumption, the authors analytically derive the optimal policy parameters (w.r.t the risk measure), as well as a general gradient estimation algorithm for both the static case (one-step decision), and the dynamic (or sequential) case.

The paper is well-written and organized. Although I am not familiar with risk-sensitive optimization, I was able to follow most of the explanations. The main contribution of this work is the unified analysis of a general class of risk measures that include most well-known measures. I believe that this is a significant contribution. However, a deeper empirical analysis of the proposed gradient algorithm would be appreciated.

Some comments/questions 1- The preliminaries section is not clear enough. I was confused about the roles of P_{\theta} and \ksi . 2- Isn't Equation 2 just the definition of a convex envelope? Why is this more restrictive than Theorem 2.1? 3- I would suggest adding a comment on how the analysis generalize to distributions on continuous variables. 4- It would be nice to mention in the short paper the role of the smoothness of the constraint functions in proving Theorem 4.2
Summary: A strong theory paper that provides a general approach to risk-sensitive decision-making. Results seem compelling but proofs need to be verified.

Author Feedback
Author rebuttal: We thank the reviewers for their comments. We will incorporate all the suggestions in the paper.

Reviewer 1:
Our sample-based policy gradient algorithm in Section 4.3 is based on the sampled average approximation method (SAA), for which in general, and under mild technical conditions, a law-of-large-numbers (LLN) style variance bound of 1/sqrt(number of samples) can be guaranteed [26]. While we have not explicitly extended this result for the case in the paper, we expect that under some technical conditions, a similar result should hold. This bound is comparable to the variance bound that can be obtained with the analytical-solution based policy-gradient methods, which are also based on the LLN. Thus, at least asymptotically, we expect both methods to be comparable.

We agree that an empirical evaluation is important. Therefore, as the reviewer suggested, we performed a comparison between the sample-based policy gradient and the analytical-solution based one for CVaR. The results corroborate our expectations and suggest that the two approaches are very similar in performance. The results may be accessed in the (anonymous) URL: https://sites.google.com/site/coherentrisknips15rebuttal/
We will add this important discussion to the paper.

Reviewer 3:
Eq. (2) states that the risk envelope can be written in a standard convex programming framework. This framework is general enough to capture almost all commonly used risk metrics such as the CVaR and spectral risk measures, and allows us to systematically analyze the Lagrange multipliers and solution vector required in deriving the policy gradient. However, by definition, this is still more restrictive than Thm 2.1, which only says that the problem is convex. For example in "Vector-valued Coherent Risk Measures" by E. Jouini, M. Meddeb, and N. Touzi 2007, the risk envelope described has a set-valued constraint. Such generalization is out of the scope of our paper.

Reviewer 5:
Our actor-critic algorithm requires a simulator. We will clarify this point in the paper.